

# Towards the one loop IR/UV dictionary in the SMEFT: One loop generated operators from new scalars and fermions

Guilherme Guedes[1*], Pablo Olgoso[2†] and José Santiago[2‡]

**1** Deutsches Elektronen-Synchrotron DESY, Notkestr. 85, 22607 Hamburg, Germany
**2** CAFPE and Departamento de Física Teórica y del Cosmos, Universidad de Granada, Campus de Fuentenueva, E-18071 Granada, Spain

★ guilherme.guedes@desy.de , † pablolgoso@ugr.es , ‡ jsantiago@ugr.es ,

## Abstract

Effective field theories offer a rationale to classify new physics models based on the size of their contribution to the effective Lagrangian, and therefore to experimental observables. A complete classification can be obtained, at a fixed order in perturbation theory, in the form of IR/UV dictionaries. We report on the first step towards the calculation of the one loop, dimension 6 IR/UV dictionary in the SMEFT. We consider dimension-six operators in the SMEFT that cannot be generated at tree level in weakly coupled extensions of the Standard Model. This includes operators with three gauge field strength tensors, operators with two field strength tensors and two scalar fields and dipole operators. We provide a complete classification of renormalizable extensions of the Standard Model with new scalar and fermion fields that contribute to these operators at one loop order, together with their explicit contribution. Our results are encoded in a `Mathematica` package called `SOLD` (SMEFT One Loop Dictionary), which includes further functionalities to facilitate the calculation of the complete tree level and one loop matching of any relevant model via an automated interface to `matchmakereft`. All operators in our list are indeed generated at the one loop order in the extensions considered with the exception of CP-violating ones with three field-strength tensors.

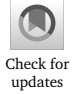

# 1  A new guiding principle in the search of physics beyond the Standard Model

During a glorious first decade of the century, the naturalness principle guided theorists into a model-building frenzy. An ever growing list of interesting new physics models was readily waiting for the Large Hadron Collider (LHC) to find them. An extra decade of intensive experimental scrutiny, by the LHC and other experiments, has shown that new physics is not likely to be as close around the corner as we expected. Indeed, there seems to be a mass gap between the scales we are experimentally probing and the scale of new physics. When this is the case effective field theory (EFT) becomes the best tool to analyse the phenomenological implications of experimental data on models of new physics.

The way EFT facilitates this analysis is by splitting the relevant calculations in two independent steps. In the bottom-up approach experimental data is parametrised in a very convenient, mostly model-independent way by means of global fits to a fixed EFT. In the second step, the top-down approach, specific new physics ultraviolet (UV) models are matched onto the EFT, thus connecting, via the global fits, theoretical models to experimental data. The matching calculation, which has to be repeated for each new physics model, can be easily performed thanks to available automated matching tools like `matchmakereft` [1] or `Matchete` [2] (see also [3]).

While naturalness arguments can still play a relevant role in the discovery and interpretation of new physics beyond the Standard Model (SM), it is clear that it cannot be our only guiding principle. The pressure from more and more stringent direct limits on the scale of new physics together with the vast number of models to test indicates that we should find alternative guiding principles in our quest to discover physics beyond the SM. It is in this respect that EFT shows its real power. Perturbation theory and power counting arguments provide an ordering principle that allows us to estimate the size of the different contributions of an EFT. Arguments based on symmetry and the topology of the different diagrams then allow us to completely classify new physics models that contribute up to a certain observable order in the EFT expansion. This classification leads to the idea of infrared (IR)/UV dictionaries, which comprise a complete classification of new physics models that contribute to the EFT at a certain loop and mass dimension order, together with the corresponding matching calculation at that particular order. The leading, tree level and mass dimension 6, IR/UV dictionary for the Standard Model effective field theory (SMEFT) was published in [4], building on previous partial calculations [5–8] (see [9] for an alternative way of constructing the dictionary).

IR/UV dictionaries have the potential to become a major guiding principle in the search of physics beyond the SM. They provide a complete list of all models (and only those) that can be experimentally accessible, including all possible correlations between different experimental observables and, even more strikingly, including even models that have never been thought of by theorists. Of course, this requires that the dictionaries be computed up to the relevant order in perturbation theory to match the experimental precision. The tree level dictionary, while extremely useful for sizeable effects, runs short when more precise experimental measurements are considered and must be extended to the one loop order. In this article we will report on the first steps towards the calculation of the complete one loop and mass dimension 6 IR/UV dictionary for the SMEFT.

Given the significant challenges inherent to extending the dictionary to the one loop level, we will restrict ourselves in this work to a subset of dimension 6 operators, namely those that cannot be generated at tree level in any weakly coupled extension of the SM [10, 11].[1] We will also consider extensions of the SM with an arbitrary number of new scalar and fermion fields and renormalisable interactions. Completing the dictionary will require adding new heavy vectors, non-renormalisable interactions and extending the classification to all operators in a physical basis of the SMEFT at dimension 6. We plan to do this in future works.

The rest of the article is organised as follows. We describe the operators considered in this work and their UV origin in Section 2. The matching procedure used to complete the classification and to obtain the corresponding result for the Wilson coefficients (WC) is discussed in Section 3. The partial result of the dictionary computed in this work is presented in Section 4 in the form of the `Mathematica` package SOLD, followed by an example of the usage of the dictionary for phenomenological studies in Section 5. We conclude in Section 6 and leave some technical particular results for Appendices A, B and C.

## 2 One loop generated operators in the SMEFT: Classification and UV origin

Given an EFT, the SMEFT at mass dimension 6 in our case, one can match any new physics model in several different ways. Our approach is to perform a diagrammatic off-shell matching which requires the definition of a physical basis and a Green's basis. The former, for which we adopt the Warsaw basis [18], is all we need to compute physical observables. The latter, for which we follow `matchmakereft`'s convention [1], is needed to perform the off-shell matching, which amounts to equating the tree level off-shell amplitudes in the EFT to the hard region contribution of the one-light-particle-irreducible (1lPI) amplitudes in the UV model to the required order in perturbation theory. Redundant and evanescent[2] operators in the Green's basis are then reduced to the ones in the physical basis. Thus, considering the UV origin of a specific operator in the Warsaw basis requires including the contribution to all redundant and evanescent operators that contribute to it.

Any operator in an EFT can be generated at different orders in perturbation theory, depending on the UV model considered. Certain operators, however, can never be generated at tree level in any weakly coupled extension of the SM. This was shown, using simple topological arguments, in [10] for the SMEFT at dimension 6 in a basis along the lines of what many years

---

[1]See [12] for a related analysis and [13, 14] for recent efforts towards the calculation of the dictionary for four-fermion interactions. See also [15–17] for results on generic UV extensions for a set of operators in the low energy EFT (LEFT).

[2]We follow in this work the slightly unconventional nomenclature of `matchmakereft` by defining evanescent operators as those that are *equivalent* to operators in the physical basis in $d = 4$ space-time dimensions rather than the more traditional one in which they are defined to be *vanishing* in $d = 4$.

Table 1: one loop generated operators in the Warsaw basis. Shaded operators are generated at two or higher order loops in SM extensions with fermions and scalars.

| $X^3$ | $X^2H^2$ | $\psi^2XH$ + h.c. |
|---|---|---|
| $\mathcal{O}_{3G} = f^{ABC}G_\mu^{A\nu}G_\nu^{B\rho}G_\rho^{C\mu}$ | $\mathcal{O}_{HG} = G_{\mu\nu}^A G^{A\mu\nu}H^\dagger H$ | $\mathcal{O}_{uG} = (\bar{q}T^A\sigma^{\mu\nu}u)\widetilde{H}G_{\mu\nu}^A$ |
| $\mathcal{O}_{3\widetilde{G}} = f^{ABC}\widetilde{G}_\mu^{A\nu}G_\nu^{B\rho}G_\rho^{C\mu}$ | $\mathcal{O}_{H\widetilde{G}} = \widetilde{G}_{\mu\nu}^A G^{A\mu\nu}H^\dagger H$ | $\mathcal{O}_{uW} = (\bar{q}\sigma^{\mu\nu}u)\sigma^I\widetilde{H}W_{\mu\nu}^I$ |
| $\mathcal{O}_{3W} = \epsilon^{IJK}W_\mu^{I\nu}W_\nu^{J\rho}W_\rho^{K\mu}$ | $\mathcal{O}_{HW} = W_{\mu\nu}^I W^{I\mu\nu}H^\dagger H$ | $\mathcal{O}_{uB} = (\bar{q}\sigma^{\mu\nu}u)\widetilde{H}B_{\mu\nu}$ |
| $\mathcal{O}_{3\widetilde{W}} = \epsilon^{IJK}\widetilde{W}_\mu^{I\nu}W_\nu^{J\rho}W_\rho^{K\mu}$ | $\mathcal{O}_{H\widetilde{W}} = \widetilde{W}_{\mu\nu}^I W^{I\mu\nu}H^\dagger H$ | $\mathcal{O}_{dG} = (\bar{q}T^A\sigma^{\mu\nu}d)HG_{\mu\nu}^A$ |
| | $\mathcal{O}_{HB} = B_{\mu\nu}B^{\mu\nu}H^\dagger H$ | $\mathcal{O}_{dW} = (\bar{q}\sigma^{\mu\nu}d)\sigma^IHW_{\mu\nu}^I$ |
| | $\mathcal{O}_{H\widetilde{B}} = \widetilde{B}_{\mu\nu}B^{\mu\nu}H^\dagger H$ | $\mathcal{O}_{dB} = (\bar{q}\sigma^{\mu\nu}d)HB_{\mu\nu}$ |
| | $\mathcal{O}_{HWB} = W_{\mu\nu}^I B^{\mu\nu}H^\dagger\sigma^I H$ | $\mathcal{O}_{eW} = (\bar{\ell}\sigma^{\mu\nu}e)\sigma^IHW_{\mu\nu}^I$ |
| | $\mathcal{O}_{H\widetilde{W}B} = \widetilde{W}_{\mu\nu}^I B^{\mu\nu}H^\dagger\sigma^I H$ | $\mathcal{O}_{eB} = (\bar{\ell}\sigma^{\mu\nu}e)HB_{\mu\nu}$ |

later would become the Warsaw basis, including the redundant operators that contribute to the ones in the physical basis. The complete list of operators in the Warsaw basis that cannot be generated at tree level are grouped in three classes. These are operators with three field strength tensors, class $X^3$, operators with two field strength tensors and two Higgs bosons, class $X^2H^2$ and finally dipole operators in class $\psi^2XH$. They are all collected in Table 1.

Let us discuss one loop contributions to each operator class in turn. In particular, we are interested in the UV origin of such contributions, namely, which heavy scalar or fermion fields can give rise to these operators at one loop order. The fact that light and heavy fields are in complete, independent representations of the SM gauge group means that any gauge boson insertion does not change the type of field in the amplitude. Thus, we do not need to worry about gauge boson insertions in the following discussion (of course all required gauge boson insertions will be properly taken into account when computing the actual amplitudes). Operators in classes $X^3$ and $X^2H^2$ do not receive contributions from redundant or evanescent operators so we only need to focus on the generation of the physical operators. Operators in the former class, $X^3$, can be computed from three gauge boson off-shell amplitudes. Neglecting the gauge boson insertions these amplitudes are simply vacuum bubbles of a single heavy scalar or fermion that is charged under the corresponding gauge group. Thus, any non-singlet scalar or fermion multiplet will, in principle, contribute to the corresponding operator. Operators in the $X^2H^2$ class can be computed from $H^\dagger H$ plus two gauge boson off-shell amplitudes. Taking into account again that gauge boson vertices do not change the matter content, these can be computed simply from $H^\dagger H$ two-point functions (dressed with insertions of two gauge bosons). Dipole operators, on the other hand, receive contributions also from redundant and evanescent operators. All the relevant operators, including redundant and evanescent ones (see Appendix C for a complete list) can be computed from amplitudes of the form $\bar{\psi}\psi(V)$ and $\bar{\psi}_L\psi_R H^{(\dagger)}(V)$, where $V$ stands for the corresponding gauge boson, $\psi_L = \{l_L, q_L\}$ stand for the left-handed fermions of the SM, $\psi_R = \{u_R, d_R, e_R\}$ for the right-handed ones and $\psi = \{\psi_L, \psi_R\}$ includes both. Neglecting again the gauge boson insertions we just need to consider $\bar{\psi}\psi$ and $\bar{\psi}_L\psi_R H^{(\dagger)}$ amplitudes.

Once we know which amplitudes we need to compute we can use topological considerations to fix the actual Feynman diagrams contributing to them. Any one loop Feynman diagram satisfies the following relation between the number of external particles, $E$, and the number of vertices,

$$E = \sum_{n=3}^{E+2}(n-2)V_n,\tag{1}$$

where $n$ denotes the order (number of fields) of the vertices and $V_n$ the number of vertices of order $n$ in the diagram and the sum ends at $n = E + 2$ because higher order vertices would result in more than $E$ external legs at one loop. In particular we are interested in the cases $E = 2, 3$, for which we have

$$E = 2 = V_3 + 2V_4 \,, \tag{2}$$

$$E = 3 = V_3 + 2V_4 + 3V_5 \,. \tag{3}$$

Using these expressions and the form of possible renormalisable vertices (which further limits $n \leq 4$) between light and heavy scalars or fermions one can draw all generic Feynman diagrams contributing to the hard region of the corresponding 1lPI amplitudes.[3] From these diagrams one can immediately determine the quantum numbers of possible UV completions of the operators we are interested in.[4] In practice, we can also determine the quantum numbers of possible UV completions *a posteriori,* by using the explicit result of the matching to a generic UV model as we describe in detail in the next section. We have cross-checked both ways of classifying UV models finding full agreement in all cases.

All the operators in Table 1 are indeed generated at one loop order in extensions of the SM with heavy scalars and/or fermions with the notable exception of the CP-violating operators in the $X^3$ class, namely $\mathcal{O}_{\widetilde{3W}}$ and $\mathcal{O}_{\widetilde{3G}}$, that can only be generated at the two loop order in these models. This feature was already pointed out, within some simplifying assumptions, in [20].

## 3 Computing the implications of new physics models

### 3.1 Matching procedure

Once we have fixed the list of possible new physics models that contribute at one loop order to a certain operator (in the Green's or physical basis) we have to compute their explicit contribution to the corresponding WC. Before explaining how we have performed this calculation in general, let us emphasize that the operators we are interested in are first generated at the one loop order. This means that we do not need to keep track of "universal" contributions in the form of wave-function renormalization or one loop contributions to the renormalizable couplings, that would give additional contributions to the one loop dictionary only via tree level generated operators. Thus, tree level generated operators and one loop contribution to the renormalizable couplings (including kinetic terms) will be disregarded in the following. The only exception to this is the possible tree level generation of evanescent operators that, via Fierzing, can give rise to one loop rational contribution to the dipole operators we are interested in (see Refs. [21,22] for a recent detailed discussion). Given the small number of extensions in which this effect is relevant we provide the complete list in Appendix B.

We use `matchmakereft` to perform the matching, which, as mentioned above, follows a diagrammatic off-shell approach. Since we do not know *a priori* the quantum numbers of the heavy particles we have proceeded in two steps. First we have defined a generic model consisting of an extension of the SM with heavy (Dirac or Majorana) fermions and (real or complex) scalars with the most general couplings, among themselves and with the SM particles, as allowed by Lorentz invariance but leaving their gauge quantum numbers, and therefore hthe

---

[3]In particular this means that diagrams with light bridges or loops involving only light particles are not to be considered.

[4]See [19,20] for a similar approach to classifying UV completions of the SMEFT with some simplifying assumptions.

corresponding Clebsch-Gordan (CG) coefficients, arbitrary. The only assumptions made on this generic model are the following:

- It respects the SM gauge group symmetry, under which the new heavy particles transform in some arbitrary representation.

- The interaction basis coincides with the mass-eigenstates basis, i.e., mass terms are diagonal at tree level.

- Heavy fermions are vector-like, i.e., both chiralities transform under the same representation of the gauge group, so the UV theory has no chiral anomaly.

Denoting all fermions (light and heavy) by a single field $\Psi_a$ and all scalars (light and heavy) by another one $\Phi_b$, where the indices, $a, b$, run over all relevant multiplets under the SM gauge symmetry, we can write the generic form of the Lagrangian as follows

$$
\begin{aligned}
\mathcal{L}_{\text{UV}} = {}& \delta_{\Psi_a} \bar{\Psi}_a \Big[ i\slashed{D} - M_{\Psi_a} \Big] \Psi_a + \delta_{\Phi_a} \Big[ |D_\mu \Phi_a|^2 - M_{\Phi_a}^2 |\Phi_a|^2 \Big] \\
&+ \sum_{\chi=L,R} \Big[ Y^\chi_{abc} \overline{\Psi}_a P_\chi \Psi_b \Phi_c + \widetilde{Y}^\chi_{abc} \overline{\Psi}_a P_\chi \Psi_b \Phi_c^\dagger + X^\chi_{abc} \overline{\Psi^c}_a P_\chi \Psi_b \Phi_c + \widetilde{X}^\chi_{abc} \overline{\Psi^c}_a P_\chi \Psi_b \Phi_c^\dagger + \text{h.c.} \Big] \\
&+ \Big[ \kappa_{abc} \Phi_a \Phi_b \Phi_c + \kappa'_{abc} \Phi_a \Phi_b \Phi_c^\dagger + \lambda_{abcd} \Phi_a \Phi_b \Phi_c \Phi_d \\
&\qquad + \lambda'_{abcd} \Phi_a \Phi_b \Phi_c \Phi_d^\dagger + \lambda''_{abcd} \Phi_a \Phi_b \Phi_c^\dagger \Phi_d^\dagger + \text{h.c.} \Big],
\end{aligned}
\tag{4}
$$

where $P_{L,R} = (1 \mp \gamma^5)/2$ are the usual chirality projectors, $\Psi^c \equiv \mathcal{C}\overline{\Psi}^T$ with $\mathcal{C}$ the charge conjugation matrix, $\delta_{\Psi_a}$ is 1 (1/2) times the identity matrix for complex (Majorana, satisfying $\Psi_a^c = \Psi_a$) fermions and $\delta_{\Phi_a}$ is 1 (1/2) times the identity matrix for complex (real, satisfying $\Phi_a^\dagger = \Phi_a$) scalars and the masses are zero for all the light fields except for the SM Higgs doublet. The remaining couplings represent, for each fixed value of the indices, coupling constants times CG tensors. Our convention for the covariant derivative is the following:

$$
D_\mu \Psi = (\partial_\mu - i g_1 B_\mu \mathbf{Y}_\Psi - i g_2 W_\mu^a T_W^a - i g_3 G_\mu^a T_G^a) \Psi,
\tag{5}
$$

where $T_W (T_G)$ are the generators of SU(2) (SU(3)) in the representation of $\Psi$, and $\mathbf{Y}_\Psi$ is its hypercharge. Also note that, despite the explicit sign for the interactions in Eq.(4), we follow the standard convention for the SM interactions so that

$$
\widetilde{Y}^R_{q^i u^j \phi} = -i\sigma^2 (Y_u)_{ij},
\tag{6}
$$

$$
Y^R_{q^i d^j \phi} = -(Y_d)_{ij},
\tag{7}
$$

$$
Y^R_{l^i e^j \phi} = -(Y_e)_{ij},
\tag{8}
$$

$$
\lambda''_{\phi^4} = -\lambda/2,
\tag{9}
$$

with $\sigma^2$ the second Pauli matrix. Once the generic UV model has been defined we have used `matchmakereft` to perform the calculation of the amplitudes, their expansion in the hard region and the projection over kinematic configurations and we have solved for the WCs of the different operators. The gauge information is however still left generic at this point. This result is stored internally, as it will be unchanged for any extension of the SM and the final user does not need to repeat this calculation. In a second step, once the specific gauge representations for the heavy fields are fixed, we use `GroupMath` [23] to perform the remaining group-theoretic calculation. In order to obtain the results in the Warsaw basis we have computed all the relevant coefficients in the Green's basis and the result has then been translated to the physical basis using the redundancies provided in [1]. We follow the naive dimensional regularization prescription for $\gamma^5$, as implemented in `matchmakereft`, which is compatible with the scheme introduced in [22] to compute the evanescent contributions reported in Appendix B.

## 3.2 Model classification

The bottom-up use of the dictionary consists of the classification of all possible renormalisable SM extensions (including heavy fermion and scalar fields) whose one loop contribution to a certain WC, either in the SMEFT Warsaw or Green's basis, is allowed by gauge symmetry. This classification can be given in a closed form, even if the number of possible models is infinite. The reason is that, contrary to what happens in the tree level case, quadratic couplings in heavy fields can contribute for the first time at one loop order. This allows for loop topologies in which the gauge representations for the fields running in the loop are not fixed, but only their product is. As a consequence, one can only impose restrictions for the fields to contribute through a certain diagram, but those can be fulfilled by an infinite number of representations. Thus, the classification of possible new physics models can be given at two different levels: on the first level we provide a complete, finite list of the restrictions to be fulfilled by the new fields; on a second level we give a list of the allowed specific representations that satisfy any of these conditions, up to certain dimension of such representations (the list being infinite otherwise). As we will see below, the `Mathematica` package SOLD, that encodes the one loop dictionary, includes routines to perform both tasks.

The list of restrictions (first level) can be easily computed in a comprehensive way using the intermediate results of the matching as discussed in the previous section. Once we have performed the matching for a specific WC in terms of a combination of CG tensors, we can simply check the restrictions on the quantum numbers of the heavy fields so that each diagram is allowed by the gauge symmetry. Note however that we are just imposing that the result is non-zero *a priori*; the particular value of the gauge structure depends on specific choices for the representations, so it could happen that it vanishes for some of them, or even that some cancellation happens between different diagrams. The list of restrictions for each diagram defines implicitly a possible new extension and it is added to the complete list. These restrictions are then reduced so that they contain the minimum number of different fields needed to satisfy them. Finally, we eliminate from the complete list those models that are related by conjugation of one of the fields, since they are physically equivalent. In the case of coefficients in the Warsaw basis, we compute this list for every coefficient that can contribute to it through redundancies.

The complete list of models, even at the first level, is too long to report here and is given in electronic form via the SOLD package. The only exception is the operators in the $X^3$ class, for which both the classification and the result can be given in closed form. We report on the results for this class on Appendix A. An interesting result of our calculation is that the two CP-violating operators with three field strength tensors, namely, $\mathcal{O}_{\widetilde{3W}}$ and $\mathcal{O}_{\widetilde{3G}}$, are not generated at the one loop order in any renormalizable extension of the SM with heavy scalars or fermions.

# 4 SOLD **usage**

We provide in this section a detailed description of the `Mathematica` package SOLD, that encodes the calculation of the part of the SMEFT one loop dictionary as described in this article.

## 4.1 Installation

SOLD is publicly available in the following Gitlab repository: https://gitlab.com/jsantiago_ugr/sold. Before installing SOLD, the user should make sure that both `GroupMath` and

```
In[1]:=  << SOLD`
```

SMEFT One Loop Dictionary loaded
Version: 1.0.1
Authors: Guilherme Guedes, Pablo Olgoso, José Santiago
Reference: arXiv:2303.16965
Webpage: https://gitlab.com/jsantiago_ugr/sold

```
XXXXXXXXXXXXXXXXXXXXXXXXXXX GroupMath XXXXXXXXXXXXXXXXXXXXXXXXXXX
Version: 1.1.2 (6/May/2020)
Author: Renato Fonseca
Reference: 2011.01764 [hep-th]
Website: renatofonseca.net/groupmath
Built-in documentation: here
XXXXXXXXXXXXXXXXXXXXXXXXXXXXXXXXXXXXXXXXXXXXXXXXXXXXXXXXXXXXXX-
    XXXX
```

Figure 1: Loading SOLD.

`matchmakereft` are already installed.[5] There are two ways of installing the package:

1. **Automatic installation**. SOLD can be installed in a fully automated way by typing the following command on a `Mathematica` notebook:

   ```
   In[1]:=  Import["https://gitlab.com/jsantiago_ugr/sold/-/raw/main
           /install.m"]
   ```

   This will download the package and place it in the `Applications` folder of `Mathematica`'s base directory. The same command will (re)install the latest version available in the repository.

2. **Manual installation**. The alternative way is to manually download the package from the SOLD repository and place it in the `Applications` folder of `Mathematica`'s base directory, or a different directory as long as it is included in the variable $Path.

   Once installed, SOLD can be loaded in any `Mathematica` notebook in the usual way

   ```
   In[2]:=  << SOLD`
   ```

with an output shown in Fig. 1.

## 4.2   List of functions

The following functions are available in the SOLD package. The usual help command in `Mathematica` can be used to obtain more information on them.[6] An updated version of the manual can be found in SOLD's installation directory.

- `OneLoopOperatorsGrid`. Displays a grid with the SMEFT operators in the Warsaw basis whose leading contribution is at one loop. When the mouse is on top of each entry the expression of the operator is displayed, and when clicked, the different contributions from coefficients of the Green's basis are shown.

---

[5]Technically `matchmakereft` is not necessary if the user is only interested in the operators described in this article. It is necessary if the full one loop matching (including operators not in the three classes studied here) is required.

[6]We provide detailed examples of the output for the most relevant functions in the next section.

- `ListModelsWarsaw[coefficient]`. Returns a list with all possible SM extensions (sometimes implicitly defined by restrictions in the product of some representations) whose contribution to `coefficient` in the SMEFT Warsaw basis is allowed by gauge symmetry (the conventions for the coefficients follow `matchmakereft` and the list of coefficients is stored in the variable `AllCoefficientsWarsaw`). Each entry of the result represents a different SM extension. For each entry of the list, the first item indicates the field content of the model (number and spin of heavy fields, with $\phi i$ and $\psi i$ indicating scalars and fermions, respectively, with $i$ a number), the second item contains the restrictions that the $SU(3) \times SU(2)$ representations of the new fields should fulfill, and the last item indicates the hypercharge restrictions.

- `ListModelsGreen[coefficient]`. Identical to the previous function but for operators in the Green's basis.

- `ListValidQNs[listrestrictions,<MaxDimSU3>,<MaxDimSU2>]`. Computes the valid representations under $SU(3) \times SU(2)$ representations, up to dimensions `MaxDimSU3` and `MaxDimSU2`, respectively, allowed by `listrestrictions`, for the fields contained in it. `listrestrictions` can be either the direct output of `ListModelsWarsaw` or `ListModelsGreen`, a sublist of its entries or just an entry's second item. `MaxDimSU3` and `MaxDimSU2` are optional arguments and their default values are 15 and 5, respectively.

- `Match2Warsaw[coefficient, extension]`. Computes the contribution to a particular WC `coefficient` in the Warsaw basis generated by a model defined by `extension`, where `extension` is a list of replacement rules with a tag to identify the heavy particle (that must begin with an S or F, depending on whether the heavy particle is a scalar or a fermion respectively and be followed by an identifying letter) and a list of its quantum numbers under $SU(3) \times SU(2) \times U(1)$. As an explicit example, if the user is interested in calculating the matching conditions of $(\mathcal{O}_{eW})_{i,j}$ from a full theory with an $SU(2)$ triplet vector-like lepton of hypercharge -1, a triplet scalar leptoquark of hypercharge -1/3 and a triplet vector-like quark of hypercharge -4/3 [24], they would need to write:

```
In[3]:= Match2Warsaw[alphaOeW[i,j],
           {Sa->{3,3,-1/3},Fa->{1,3,-1},Fb->{3,3,-4/3}}]
```

Note that the definitions of `coefficient` follow `matchmakereft` convention, in particular `iCPV` fixes the Levi-Civita convention via $iCPV = \epsilon_{0123}$. Explicit numerical values for the flavour indices are also allowed.

In order to allow for different, non-equivalent $SU(3)$ representations of the same dimension, as well as conjugated ones, Dynkin indices can be used as a valid input for $SU(3)$. Note that this is not necessary for $SU(2)$. An explicit example of this format is provided in the next section. Symbolic hypercharges for the heavy fields are also supported, as long as they are called `Yi`, where `i` is an integer character. However, only vertices that formally conserve hypercharge will be taken as non-zero. This means, for instance, that fields with symbolic hypercharge will never couple linearly with SM.

Finally, we consider fermions to be Majorana if they transform in a real representation and Dirac otherwise. In order to obtain the results for a Dirac field in a real representation, the user should use two degenerate Majorana fields to reproduce this case.

- `Match2Green[coefficient,extension]`. Computes the contribution to `coefficient` in the Green's basis, defined in [1]. The conventions for `coefficient` and `extension` are the same as for the function `Match2Warsaw`.

- `NiceOutput[result,<ListSubstitutions>]`. Returns a more readable expression of `result`. `ListSubstitutions` is optional and set to `False` by default; if set to `True`, prints a list of the substitutions performed.

- `SOLDInputForm[fieldreps]`. Translates the gauge representation of a field (including its hypercharge) from the output form given by `ListValidQNs` to a valid input form usable by `Match2Warsaw`, `Match2Green`, `CreateLag`, `GenerateMMEModel` or `CompleteOneLoopMatching`. An explicit example of this function is given in the next section.

- `CreateLag[extension]`. Returns the full (BSM) Lagrangian internally used to compute results produced by `extension`, including the numerical values of each of the CG tensors, which are presented as `TSi` or `TCi` for the SU(2) and SU(3) contraction respectively, where `i` corresponds to an identifying number.

- `GenerateMMEModel[extension, modelname, <outputdirectory>]`. Generates, in the `outputdirectory`, the `matchmakereft` model needed for the full one loop computation (the files included are useful to use with other tools such as `FeynRules`). The `modelname.fr` file contains the full Lagrangian of `extension`, the heavy particle definitions and parameter definitions, all in `FeynRules` format. The file `SM_SOLD.fr` contains the SM definition in `FeynRules` format and in case the heavy particles have exotic representations under the SM gauge groups, it adds these representations to the definition of the gauge groups. The file `modelname.gauge` contains the numerical definitions of the CG tensors considered in the definition of the Lagrangian. `outputdirectory` is optional and set to `Mathematica`'s current working directory by default.

- `CompleteOneLoopMatching[extension, modelname, <EFTname>, <outputdirectory>]`. Runs `matchmakereft` to obtain the complete one loop matching conditions between a UV `extension` and an effective theory `EFTname`. If there is no `modelname_MM` in `outputdirectory`, `GenerateMMEModel` is called in the first place. `EFTname` is an optional argument and takes the default value of the `matchmakereft`'s model for the SMEFT, `SMEFT_Green_BPreserving_MM`. `outputdirectory` is also optional and set to `Mathematica`'s current working directory by default.[7]

Note that `matchmakereft` must be installed to run these last two functions, `GenerateMMEModel` and `CompleteOneLoopMatching`.

## 4.3 Example of usage

In this subsection we will show an example of the usage of the package functions in sequential order. As a matter of example, and in preparation for the phenomenological study in the next section, let us consider that the user is interested in exploring UV completions which could generate the SMEFT operator $\mathcal{O}_{dG}$.

After loading SOLD we can start by listing the conditions on models that generate this operator in the Warsaw basis. The corresponding command is

In[4]:= `ListModelsWarsaw[alphaOdG[i,j]]`

whose output is partially shown in Fig. 2. Note that, for the SU(3) × SU(2) quantum numbers a rule means that the representation for the corresponding particle is fixed whereas when the

---

[7]When run from a `Mathematica` notebook `matchmakereft` prints the output only at the end of the full run. For a more informative output we recommend the user to create the model within SOLD using the `GenerateMMEModel` function but then run the matching from `matchmakereft` in the terminal directly.

```
In[2]:= listofmodels = ListModelsWarsaw[alphaOdG[i, j]];
      MatrixForm[Join[Take[listofmodels[[1]], {1, 3}], {{"....", "....", "...."}}, Take[listofmodels[[1]], {20, 22}],
       {{"....", "....", "...."}}, Take[listofmodels[[1]], {145, 146}], {{"....", "....", "...."}}]]
```

Out[3]//MatrixForm=

$$
\begin{pmatrix}
\text{Field Content} & SU(3) \otimes SU(2) & U(1) \\
\{\phi 1\} & \{\phi 1 \to \mathbf{\bar{3}} \otimes \mathbf{1}\} & \{Y_{\phi 1} \to \frac{1}{3}\} \\
\{\phi 1\} & \{\phi 1 \to \mathbf{\bar{3}} \otimes \mathbf{1}\} & \{Y_{\phi 1} \to \frac{4}{3}\} \\
\dots & \dots & \dots \\
\{\phi 1, \phi 2\} & \{\phi 1 \to \mathbf{8} \otimes \mathbf{2}, \phi 2 \otimes \overline{\phi 2} \supset \mathbf{8} \otimes \mathbf{3}\} & \{Y_{\phi 1} \to -\frac{1}{2}, Y_{\phi 2}\} \\
\{\phi 1, \psi 1\} & \{\psi 1 \otimes \overline{\phi 1} \supset \mathbf{\bar{3}} \otimes \mathbf{1}\} & \{Y_{\psi 1} \to \frac{1}{3} + Y_{\phi 1}\} \\
\{\phi 1, \psi 1\} & \{\psi 1 \otimes \overline{\phi 1} \supset \mathbf{\bar{3}} \otimes \mathbf{2}\} & \{Y_{\psi 1} \to -\frac{1}{6} + Y_{\phi 1}\} \\
\dots & \dots & \dots \\
\{\phi 1, \psi 1, \psi 2\} & \{\psi 1 \otimes \overline{\phi 1} \supset \mathbf{\bar{3}} \otimes \mathbf{2}, \psi 1 \otimes \psi 2 \supset \mathbf{1} \otimes \mathbf{2}, \psi 2 \otimes \phi 1 \supset \mathbf{\bar{3}} \otimes \mathbf{1}\} & \{Y_{\psi 1} \to -\frac{1}{6} + Y_{\phi 1}, Y_{\psi 2} \to -\frac{1}{3} - Y_{\phi 1}\} \\
\{\phi 1, \psi 1, \psi 2\} & \{\psi 1 \otimes \overline{\phi 1} \supset \mathbf{\bar{3}} \otimes \mathbf{2}, \psi 2 \otimes \overline{\psi 1} \supset \mathbf{1} \otimes \mathbf{2}, \psi 2 \otimes \overline{\phi 1} \supset \mathbf{\bar{3}} \otimes \mathbf{1}\} & \{Y_{\psi 1} \to -\frac{1}{6} + Y_{\phi 1}, Y_{\psi 2} \to \frac{1}{3} + Y_{\phi 1}\} \\
\dots & \dots & \dots
\end{pmatrix}
$$

Figure 2: Partial output of the command `ListModelsWarsaw[alphaOdG[i,j]]`.

```
In[13]:= modelQNs = ListValidQNs[listofmodels[[1, 145]]];
      Print["Model restriction :", listofmodels[[1, 145]], "\nList of Models:\n",
       MatrixForm[Join[Take[modelQNs, {1, 3}], {{"....", "....", "...."}}, Take[modelQNs, {-3, -1}]]]]
```

Model restriction : $\left\{\{\phi 1, \psi 1, \psi 2\}, \{\psi 1 \otimes \overline{\phi 1} \supset \mathbf{\bar{3}} \otimes \mathbf{2}, \psi 1 \otimes \psi 2 \supset \mathbf{1} \otimes \mathbf{2}, \psi 2 \otimes \phi 1 \supset \mathbf{\bar{3}} \otimes \mathbf{1}\}, \{Y_{\psi 1} \to -\frac{1}{6} + Y_{\phi 1}, Y_{\psi 2} \to -\frac{1}{3} - Y_{\phi 1}\}\right\}$

List of Models:

$$
\begin{pmatrix}
\phi 1 \to \mathbf{1} \otimes \mathbf{1} \otimes Y_{\phi 1} & \psi 1 \to \mathbf{\bar{3}} \otimes \mathbf{2} \otimes \left(-\frac{1}{6} + Y_{\phi 1}\right) & \psi 2 \to \mathbf{\bar{3}} \otimes \mathbf{1} \otimes \left(-\frac{1}{3} - Y_{\phi 1}\right) \\
\phi 1 \to \mathbf{1} \otimes \mathbf{2} \otimes Y_{\phi 1} & \psi 1 \to \mathbf{\bar{3}} \otimes \mathbf{1} \otimes \left(-\frac{1}{6} + Y_{\phi 1}\right) & \psi 2 \to \mathbf{\bar{3}} \otimes \mathbf{2} \otimes \left(-\frac{1}{3} - Y_{\phi 1}\right) \\
\phi 1 \to \mathbf{1} \otimes \mathbf{2} \otimes Y_{\phi 1} & \psi 1 \to \mathbf{\bar{3}} \otimes \mathbf{3} \otimes \left(-\frac{1}{6} + Y_{\phi 1}\right) & \psi 2 \to \mathbf{\bar{3}} \otimes \mathbf{2} \otimes \left(-\frac{1}{3} - Y_{\phi 1}\right) \\
\dots & \dots & \dots \\
\phi 1 \to \mathbf{15'} \otimes \mathbf{4} \otimes Y_{\phi 1} & \psi 1 \to \mathbf{10} \otimes \mathbf{3} \otimes \left(-\frac{1}{6} + Y_{\phi 1}\right) & \psi 2 \to \mathbf{\overline{10}} \otimes \mathbf{4} \otimes \left(-\frac{1}{3} - Y_{\phi 1}\right) \\
\phi 1 \to \mathbf{15'} \otimes \mathbf{4} \otimes Y_{\phi 1} & \psi 1 \to \mathbf{10} \otimes \mathbf{5} \otimes \left(-\frac{1}{6} + Y_{\phi 1}\right) & \psi 2 \to \mathbf{\overline{10}} \otimes \mathbf{4} \otimes \left(-\frac{1}{3} - Y_{\phi 1}\right) \\
\phi 1 \to \mathbf{15'} \otimes \mathbf{5} \otimes Y_{\phi 1} & \psi 1 \to \mathbf{10} \otimes \mathbf{4} \otimes \left(-\frac{1}{6} + Y_{\phi 1}\right) & \psi 2 \to \mathbf{\overline{10}} \otimes \mathbf{5} \otimes \left(-\frac{1}{3} - Y_{\phi 1}\right)
\end{pmatrix}
$$

Figure 3: Output of the command `ListValidQNs` for one restriction, that is, one entry from the output of `ListModelsWarsaw[alphaOdG[i,j]]`.

symbol $\supset$ appears only the product is constrained. For the case of U(1) an unconstrained hypercharge is explicitly written only when it does not appear in other conditions. We also include redundant restrictions such as $\phi 2 \otimes \overline{\phi 2} \supset \mathbf{1} \otimes \mathbf{1}$ because while $\phi 2$ in this case can have any quantum numbers, the information that it exists in the extension must be encoded so that the field appears when we want to find the valid quantum numbers which respect the restrictions.

After calculating the restrictions, the next step is to find the actual combinations of quantum numbers which respect them. As such the next step is to use the command

In[5]:= `ListValidQNs[conditions]`

where `conditions` stands for the output of the `ListModelsWarsaw[...]` command or a sublist of it. A fraction of the resulting list of models, using as condition the second one from the bottom appearing in Fig. 2, is shown in Fig. 3; the output is given in the format of a list where each entry corresponds to a restriction given by the previous command.

From the list of possible extensions, let us suppose the user is particularly interested in studying the first one appearing in Fig. 3, which consists of one heavy scalar and two heavy fermions with the following quantum numbers: $\phi_a \sim (1, 1, Y1)$; $\psi_a \sim (\bar{3}, 2, Y1 - 1/6)$ and $\psi_b \sim (3, 1, -Y1 - 1/3)$. The complete one loop result for the WC of the $(\mathcal{O}_{dG})_{ij}$ operator in this model is simply obtained using the following command (note the Dynkin index notation

```
In[2]:= Limit[
          Match2Warsaw[alphaOdG[i, j], {Sa → {{0, 0}, 1, Y1}, Fa → {{0, 1}, 2, -(1/6) + Y1},
              Fb → {{1, 0}, 1, -(1/3) - Y1}}] /. L1[qLbar, dR, phi][__] → 0, {MFa → MSa, MFb → MSa}] //
          FullSimplify
```

$$\text{Out[2]=} \quad -\frac{1}{384\,\text{MSa}^2\,\pi^2}\,\text{g3}\,(\text{L1}[\text{Fa, Fb, phi, L}] - 3\,\text{L1}[\text{Fa, Fb, phi, R}]) \times \text{L1bar}[\text{dRbar, Fb, Sa}][\text{j}] \times \text{L1bar}[\text{Sabar, Fa, qL}][\text{i}]$$

Figure 4: Result for the WC of the $(\mathcal{O}_{dG})_{ij}$ operator in the Warsaw basis for a particular extension in the limit of degenerate masses and neglecting terms proportional to the down-type Yukawa couplings. See the text for the precise definition of the model.

for the SU(3) representations)

```
In[6]:=   Match2Warsaw[alphaOdG[i,j],
          {Sa->{{0,0},1,Y1},Fa->{{0,1},2,-(1/6)+Y1},Fb->{{1,0},1,-(1/3)-Y1}}]
```

Incidentally, the model in the correct format such that it can be used in `Match2Warsaw` (as a list of replacement rules between the heavy particles and their quantum numbers) can be obtained directly from the output of `ListValidQNs[...]` as follows

```
In[7]:= ourModel = SOLDInputForm /@ modelQNs[[1]]
```

```
Out[7]=  {Sa->{{0,0},1,Y1},Fa->{{0,1},2,-(1/6)+Y1},Fb->{{1,0},1,-(1/3)-Y1}}
```

where `modelQNs` is defined in Fig. 3. The result from `Match2Warsaw` is given in Fig. 4, in the limit of equal masses and vanishing down Yukawa couplings.

In this output the couplings of the BSM model are defined as `Li` with `i` an identifying integer. A bar is added to this definition when the coupling corresponds to the hermitian conjugate of the respective operator. These couplings are followed by two possible sets of arguments: the first one corresponds to the fields which compose the corresponding renormalizable operator (and an $R$ or $L$ in the case of operators with two heavy fermions corresponding to a right- or left-handed projector respectively); the second set corresponds to flavour indices in case the operator contains light fermions. The number `i` identifies couplings of operators with the same field content but with different gauge contractions. The masses of the heavy fields are defined as `MX` where `X` is the tag of the BSM state. While this raw result might not be easy to read, it is the default output as to allow the user to easily make any simplifications they desire, such as the one shown in Fig. 4, where we took the down Yukawa coupling to zero.

To see a more readable expression one can make use of the `NiceOutput` function, whose output is shown in Fig. 5. In `NiceOutput` couplings are represented by $\lambda$ (or other greek letters in case there is more than one relevant operator with the same field content), except for the Yukawa couplings which are hard-coded to be written as $y$. The subscript of these couplings is given by the fields composing the corresponding operator, whereas the superscript can either be $R$ or $L$, depending on whether the operator has a right- or left-handed projector (for operators with two heavy fermions) or the flavour indices for operators with light fermions. Notice that, when set to true, the optional argument in `NiceOutput` makes it print a list of the correspondence between the couplings in both the default and the `NiceOutput` formats.

The precise definition of one coupling can be seen by calling the function `CreateLag` which outputs the full BSM Lagrangian of the UV extension. Not only are the interaction terms defined but also the numerical values used for the CG coefficients for each coupling. The Lagrangian is presented in `FeynRules` [25] notation, where the arguments of the fields correspond to their indices, and the CG tensors are named as `TCi` and `TSi` for the SU(3) and

```
In[3]:= NiceOutput[
    Limit[
        Match2Warsaw[alphaOdG[i, j], {Sa → {{0, 0}, 1, Y1}, Fa → {{0, 1}, 2, -(1/6) + Y1},
            Fb → {{1, 0}, 1, -(1/3) - Y1}}] /. L1[qLbar, dR, phi][__] → 0, {MFa → MSa, MFb → MSa}] // FullSimplify,
    True]

Out[3]= {g3 → g₃, MSa → M_Sa, L1[Fa, Fb, phi, L] → λ^[L]_Fa,Fb,φ, L1[Fa, Fb, phi, R] → λ^[R]_Fa,Fb,φ,
    L1bar[dRbar, Fb, Sa][j] → λ̄_dR̄,Fb,Sa^[j], L1bar[Sabar, Fa, qL][i] → λ̄_S̄a,Fa,qL^[i]}
```

$$
-\frac{g_3 \left(\lambda^{[L]}_{Fa,Fb,\phi} - 3\,\lambda^{[R]}_{Fa,Fb,\phi}\right) \overline{\lambda}_{\overline{dR},Fb,Sa}^{[j]}\,\overline{\lambda}_{\overline{Sa},Fa,qL}^{[i]}}{384\,\pi^2\,M^2_{Sa}}
$$

Figure 5: Result for the WC of the $(\mathcal{O}_{dG})_{ij}$ operator in the Warsaw basis for a particular extension in the limit of degenerate masses and neglecting terms proportional to the down-type Yukawa couplings. The `NiceOutput` function has been used to obtain a more readable result. The last argument, set to `True` in this example, prints a list of the replacements performed to arrive at the output. See text for the precise definition of the model.

```
In[2]:= CreateLag[{Sa → {{0, 0}, 1, Y1}, Fa → {{0, 1}, 2, -(1/6) + Y1}, Fb → {{1, 0}, 1, -(1/3) - Y1}}]

Out[2]= {Sa² Sabar² λ_S̄a,S̄a,Sa,Sa + Sa DRbar[sp1, ff0, cc0].Fb[sp1, cc1] λ_dR̄,Fb,Sa^[ff0] TC51[cc0, cc1] +
    Sa Sabar Phi[ss2] × Phibar[ss0] λ_φ̄,S̄a,φ,Sa TS11[ss0, ss2] +
    CC[Fabar[sp1, ss0, cc0]].left[Fb[sp1, cc1]] × Phi[ss2] λ^[L]_Fa,Fb,φ TC31[cc0, cc1] × TS31[ss0, ss2] +
    CC[Fabar[sp1, ss0, cc0]].right[Fb[sp1, cc1]] × Phi[ss2] λ^[R]_Fa,Fb,φ TC31[cc0, cc1] × TS31[ss0, ss2] +
    Sabar CC[Fabar[sp1, ss1, cc1]].QL[sp1, ss2, ff0, cc2] λ_S̄a,Fa,qL^[ff0] TC41[cc1, cc2] × TS41[ss1, ss2],
    {TS11 → {{1, 0}, {0, 1}}, TC31 → {{1, 0, 0}, {0, 1, 0}, {0, 0, 1}}, TS31 → {{0, -1}, {1, 0}},
    TC41 → {{1, 0, 0}, {0, 1, 0}, {0, 0, 1}}, TS41 → {{0, -1}, {1, 0}}, TC51 → {{1, 0, 0}, {0, 1, 0}, {0, 0, 1}}}}
```

Figure 6: Output of the CreateLag function.

SU(2) contractions, respectively, with `i` an identifying integer. Kinetic terms are omitted and follow the convention in Eq. (4). The explicit values for the group generators can be obtained using the routine `RepMatrices` in `GroupMath`. An example of the output of `CreateLag` is shown in Fig. 6.

Finally, one can be interested in studying the implications of this model in other operators or observables. Using `GenerateMMEModel`, the user can generate automatically a `matchmakereft` model only specifying the representations of the new heavy fields:

```
In[8]:= GenerateMMEModel[{Sa->{{0,0},1,Y1},Fa->{{0,1},2,-(1/6)+Y1},
    Fb->{{1,0},1,-(1/3)-Y1}},"model"]
```

In order to perform the complete one loop matching to the SMEFT using `matchmakereft` one simply has to run the command `CompleteOneLoopMatching`:

```
In[9]:= CompleteOneLoopMatching[{Sa->{{0,0},1,Y1},Fa->{{0,1},2,-(1/6)+Y1},
    Fb->{{1,0},1,-(1/3)-Y1}},"model"]
```

# 5 Phenomenological applications: How to use the dictionary

Our final goal when computing the one loop IR/UV dictionary is of course to use it in phenomenological applications. Even in its current partial form it can still be used to classify in a comprehensive way the origin, and the corresponding phenomenological implications in other experimental measurements, of experimental anomalies eventually reported. Since there are currently no significant confirmed anomalies we will consider, for the sake of the exposition,

a recently reported tension in different non-leptonic decays of B mesons [26]. We refer to the original article for all the relevant details and simply take at face value one of the possible explanations of this tension in terms of the following effective Lagrangian

$$\mathcal{L} = \frac{G_F}{2} \frac{g_s}{4\pi^2} m_b \sum_{q=d,s} C_{8gq}(V_{ub}V_{uq}^* + V_{cb}V_{cq}^*)\bar{q}_L \sigma^{\mu\nu} T^A b_R G_{\mu\nu}^A + \dots, \tag{10}$$

where the dots denote the hermitian conjugate and other operators not relevant for our discussion here. In the Lagrangian above, $G_F$ is the Fermi constant, $m_b$ the bottom mass, $V_{ij}$ the corresponding entries of the CKM matrix, $\sigma^{\mu\nu} = i/2[\gamma^\mu, \gamma^\nu]$ and $T^A$ are the SU(3) generators in the fundamental representation (the Gell-Mann matrices divided by 2). The reported tension among the different B-meson decays can be alleviated provided $C_{8gd}$ and $C_{8gs}$ are in the following ranges (with some correlation on the upper range for the latter)

$$0.13 \lesssim C_{8gd}(m_b) \lesssim 0.33, \quad -0.45 \lesssim C_{8gs}(m_b) \lesssim 0.03, \tag{11}$$

where the value in parenthesis is to remind us that these correspond to renormalized WC at the scale $\mu = m_b$.

Our goal is to completely classify all possible extensions of the SM (with new scalars or fermions) that can generate these non-vanishing values up to one loop order. This can be done by first expressing the WC $C_{8gq}$ in terms of the corresponding WC in the LEFT, then using the LEFT one loop RGEs, computed in [27] to express them in terms of the relevant LEFT WC at the matching scale with the SMEFT. Using the one loop matching between the SMEFT and the LEFT, computed in [28], we can express them in terms of the SMEFT WC at the electroweak scale. Finally, using the SMEFT RGEs, computed in [29–31], we can express them in terms of the SMEFT WC at the cut-off scale whose values we can read off from our dictionary (all these steps can be simplified by automated tools like DSixTools [32, 33]). In this process we can take into account that certain WC in the SMEFT can only be generated at one loop order at the cut-off scale and therefore their effect via running or one loop matching is formally a two loop effect that can be disregarded.

Denoting generically the anomalous dimension of a WC $\alpha_i$

$$\dot{\alpha}_i \equiv 16\pi^2 \mu \frac{d\alpha_i}{d\mu}, \tag{12}$$

and working to the leading log approximation (fixed order one loop effects), we have

$$\alpha_i(\mu) = \alpha_i(\mu_0) + \frac{\dot{\alpha}_i(\alpha_j^{\text{tree}})}{32\pi^2} \log\left(\frac{\mu^2}{\mu_0^2}\right), \tag{13}$$

where $\dot{\alpha}_i$ can be evaluated at any scale and, as we have explicitly written, only the contribution from tree level generated WC needs to be included. Note that we follow the matchmakereft convention for the covariant derivative $D_\mu = \partial_\mu - ig\dots$, which is the opposite to the one used in the references above. Thus, we have changed the signs of the gauge couplings whenever necessary.

The corresponding effective Lagrangian in the LEFT reads

$$\mathcal{L}_{\text{LEFT}} = (L_{dG})_{ij}\bar{d}_{L\,i} \sigma^{\mu\nu} T^A d_{R\,j} G_{\mu\nu}^A + \dots, \tag{14}$$

resulting in

$$C_{8gq} = \frac{F_q}{g_s}(L_{dG})_{qb}, \tag{15}$$

where

$$F_q \equiv \left[ \frac{G_F}{8\pi^2} m_b (V_{ub}V_{uq}^* + V_{cb}V_{cq}^*) \right]^{-1} \approx \begin{cases} 1.8 \times 10^5 \; e^{0.9 i\pi} \; \text{TeV}, & [q = d], \\ 3.8 \times 10^4 \; \text{TeV}, & [q = s]. \end{cases} \tag{16}$$

The relevant part of the LEFT RGEs, in the up basis, reads

$$
\begin{aligned}
(\dot{L}_{dG})_{ij} &= -g_s \sum_{q_u=u,c} m_{q_u} \Big[ (L_{uddu}^{S1,RR})_{q_u j i q_u} - \frac{1}{6} (L_{uddu}^{S8,RR})_{q_u j i q_u} \Big] + \dots \\
&= g_s \sum_{q_u=u,c} m_{q_u} \Big[ (C_{quqd}^{(1)})_{i q_u q_u j} - \frac{1}{6} (C_{quqd}^{(8)})_{i q_u q_u j} \Big] + \dots,
\end{aligned}
\tag{17}
$$

where the dots stand for contributions that are one loop generated or receive no contribution from the SMEFT and we have used the following tree level matching between the SMEFT and the LEFT

$$(L_{uddu}^{S1,RR})_{ijkl} = -(C_{quqd}^{(1)})_{klij}, \qquad (L_{uddu}^{S8,RR})_{ijkl} = -(C_{quqd}^{(8)})_{klij}. \tag{18}$$

We can therefore write

$$
\begin{aligned}
C_{8gq} \frac{g_s}{F_q}\Big|_{\mu=m_b} &= (L_{dG})_{qb}\Big|_{\mu=m_b} \\
&= (L_{dG})_{qb}\Big|_{\mu=m_t} + \frac{g_s}{32\pi^2} \sum_{q_u=u,c} m_{q_u} \Big[ (C_{quqd}^{(1)})_{q q_u q_u b} - \frac{1}{6} (C_{quqd}^{(8)})_{q q_u q_u b} \Big] \log\left( \frac{m_b^2}{m_t^2} \right)\Big|_{\mu=\Lambda},
\end{aligned}
\tag{19}
$$

where the SMEFT WC on the last term can be evaluated already at the cut-off scale (other effects being formally of two loop order) and for later convenience we have chosen the top quark mass for the matching scale between the SMEFT and LEFT.

Using the matching of the SMEFT onto the LEFT up to one loop we have

$$
\begin{aligned}
(L_{dG})_{qb} =\; & \frac{v}{\sqrt{2}} V_{qk}^{\dagger} (C_{dG})_{kb} + \frac{g_s}{64\pi^2} F_1(x_W) \frac{V_{qt}^{\dagger} V_{tb} m_b}{v_T^2} \\
& + \frac{g_s}{36\pi^2} \left( 1 - \frac{m_W^2}{m_Z^2} \right) m_q (C_{Hd})_{qb} + \frac{g_s}{64\pi^2} F_2(x_W) m_t V_{qt}^{\dagger} (C_{Hud})_{tb} \\
& - \frac{g_s}{72\pi^2} \left( 1 + \frac{2m_W^2}{m_Z^2} \right) m_b V_{qk}^{\dagger} (C_{Hq}^{(1)})_{kl} V_{lb} \\
& - \frac{g_s}{576\pi^2} m_b \left\{ 8 \left( 7 + 2 \frac{m_W^2}{m_Z^2} \right) V_{qk}^{\dagger} (C_{Hq}^{(3)})_{kl} V_{lb} - 9 F_1(x_W) [V_{qt}^{\dagger} (C_{Hq}^{(3)})_{tk} V_{kb} + V_{qk}^{\dagger} (C_{Hq}^{(3)})_{kt} V_{tb}] \right\} \\
& + \dots,
\end{aligned}
\tag{20}
$$

where the dots stand for two loop effects and, at the order given, $v \approx 246$ GeV. This equality should be understood at a scale $\mu = m_t$ but, again, all terms but the first one can be already evaluated at the cut-off scale as any running effect will be of two loop order. We have defined

$$x_W \equiv \frac{m_W^2}{m_t^2}, \tag{21}$$

$$F_1(x) = \frac{1 - 6x + 3x^2 + 2x^3 - 6x^2 \log(x)}{(1-x)^4}, \tag{22}$$

$$F_2(x) = \frac{1 - 3x^2 + 4x^3 - 6x^2 \log(x)}{(1-x)^3}. \tag{23}$$

The last term in the first line of Eq. (20) corresponds to the SM contribution which, using the relation between the measured Fermi constant in muon decay and the Higgs vacuum expectation value (vev), $v_T$, [31]

$$\frac{1}{v_T^2} = \sqrt{2}G_F + \frac{1}{2}[(C_{ll})_{2112} + (C_{ll})_{1221}] - [(C_{Hl}^{(3)})_{11} - (C_{Hl}^{(3)})_{22}] \equiv \sqrt{2}G_F + \Delta G_F, \qquad (24)$$

gives the following new physics contribution

$$\frac{g_s}{64\pi^2}F_1(x_W)\frac{V_{qt}^\dagger V_{tb}m_b}{v_T^2} = \text{SM} - \frac{g_s}{64\pi^2}(V_{ub}V_{uq}^* + V_{cb}V_{cq}^*)F_1(x_W)m_b\Delta G_F. \qquad (25)$$

The last piece that we need is the RGE of $C_{dG}$ in the SMEFT, which reads,

$$(\dot{C}_{dG})_{qb} = g_s\left(C_{quqd}^{(1)} - \frac{1}{6}C_{quqd}^{(8)}\right)_{qklb}(Y_u^\dagger)_{kl} + \dots, \qquad (26)$$

where the dots denote, as always, terms that correspond to two loop effects. We therefore have

$$(C_{dG})_{qb}\big|_{\mu=m_t} = \left[(C_{dG})_{qb} + \frac{g_s}{32\pi^2}\left(C_{quqd}^{(1)} - \frac{1}{6}C_{quqd}^{(8)}\right)_{qklb}(Y_u^\dagger)_{kl}\log\left(\frac{m_t^2}{\Lambda^2}\right)\right]_{\mu=\Lambda} + \dots, \quad (27)$$

where, as explicitly stated, the right-hand side of this equation is evaluated at the cut-off scale, and we have followed `matchmakereft`'s convention for the Yukawa couplings.

Putting everything together we obtain the new physics contribution to be

$$
\begin{aligned}
C_{8gq}\frac{g_s}{F_q}\Big|_{\mu=m_b} &= \frac{g_s}{32\pi^2}\sum_{q_u=u,c}m_{q_u}\Big[(C_{quqd}^{(1)})_{qq_uq_ub} - \frac{1}{6}(C_{quqd}^{(8)})_{qq_uq_ub}\Big]\log\left(\frac{m_b^2}{m_t^2}\right) \\
&+ \frac{v}{\sqrt{2}}V_{qk}^\dagger(C_{dG})_{kb} + \frac{g_s}{64\pi^2}F_1(x_W)V_{qt}^\dagger V_{tb}m_b\Delta G_F \\
&+ \frac{g_s}{36\pi^2}\left(1 - \frac{m_W^2}{m_Z^2}\right)m_q(C_{Hd})_{qb} + \frac{g_s}{64\pi^2}F_2(x_W)m_t V_{qt}^\dagger(C_{Hud})_{tb} \\
&- \frac{g_s}{72\pi^2}\left(1 + \frac{2m_W^2}{m_Z^2}\right)m_b V_{qk}^\dagger(C_{Hq}^{(1)})_{kl}V_{lb} \\
&- \frac{g_s}{576\pi^2}m_b\left\{8\left(7 + 2\frac{m_W^2}{m_Z^2}\right)V_{qk}^\dagger(C_{Hq}^{(3)})_{kl}V_{lb} - 9F_1(x_W)[V_{qt}^\dagger(C_{Hq}^{(3)})_{tk}V_{kb} + V_{qk}^\dagger(C_{Hq}^{(3)})_{kt}V_{tb}]\right\} \\
&+ V_{qk}^\dagger\left[\frac{g_s}{32\pi^2}\left(C_{quqd}^{(1)} - \frac{1}{6}C_{quqd}^{(8)}\right)_{qkkb}(m_u)_k\log\left(\frac{m_t^2}{\Lambda^2}\right)\right] + \dots,
\end{aligned}
\qquad (28)
$$

where in the last line we neglected higher-loop and mass dimension effects to write the mass of the up-type quarks in terms of their Yukawa couplings and the Higgs vev. In the equation above the first line corresponds to the running in the LEFT between $\mu = m_b$ and $\mu = m_t$, the second to fifth to the matching between the SMEFT and the LEFT at $\mu = m_t$ and the last to the running between $\mu = m_t$ and the cut-off scale $\mu = \Lambda$. All the SMEFT WC on the RHS of this equation are to be evaluated at the cut-off scale and only their tree level contributions are relevant except for $C_{dG}$.

Equation (28) allows us to directly write the low-energy *measurements* of $C_{8gq}$ in terms of the SMEFT WC at the cut-off scale. This is where the dictionary can be directly used to fully classify all SM extensions (with scalars and fermions for the one loop case) that contribute to this observable. It receives contribution from tree level generated operators, that can be

directly read off from the tree level dictionary [4], and one contribution from the one loop generated WC $C_{dG}$, that we classify in this work. In order to simplify the discussion here we will check first which WCs can give a sizeable contribution to our observable. In order to do that we will take a benchmark point with

$$C_{8gd}^{\text{benchm.}} = 0.25, \qquad C_{8gs}^{\text{benchm.}} = -0.1, \tag{29}$$

allowed from the study of [26]. We will now rescale the WC with an explicit power of the cut-off

$$C = \frac{c}{\Lambda^2}, \tag{30}$$

with $c$ now a dimensionless coefficient and drop all contributions that require the relevant $c$ to be larger than 10 (to be on the conservative side) to reproduce the benchmark points. With these restrictions we find, for $\Lambda = 2$ TeV,

$$
\begin{aligned}
C_{8gd} \approx{} & [-6.9 + 2.9i] \times 10^3 \, (c_{dG})_{1,3} + [1.6 - 0.66i] \times 10^3 \, (c_{dG})_{2,3} - 65.5 \, (c_{dG})_{3,3} \\
& + [3.22 - 1.34i] \times 10^{-2} \, (c_{Hq}^{(1)})_{1,2} + [0.80 - 0.33i] \, (c_{Hq}^{(1)})_{1,3} - [0.18 - 0.08i] \, (c_{Hq}^{(1)})_{2,3} \\
& + [0.11 - 0.04i] \, (c_{Hq}^{(3)})_{1,2} + [2.38 - 0.99i] \, (c_{Hq}^{(3)})_{1,3} - [2.5 - 1.0i] \times 10^{-2} \, (c_{Hq}^{(3)})_{2,2} \\
& - [0.55 - 0.23i] \, (c_{Hq}^{(3)})_{2,3} - 0.39 \, (c_{Hud})_{3,3} + [1.23 - 0.51i] \, (c_{quqd}^{(1)})_{1,2,2,3} \\
& + 1.18 \, (c_{quqd}^{(1)})_{1,3,3,3} - [0.21 - 0.09i] \, (c_{quqd}^{(8)})_{1,2,2,3} - 0.20 \, (c_{quqd}^{(8)})_{1,3,3,3}, \tag{31} \\
C_{8gs} \approx{} & [373 + 7i] \, (c_{dG})_{1,3} + [1600 + 30i] \, (c_{dG})_{2,3} - 65.5 \, (c_{dG})_{3,3} \\
& - 0.043 \, (c_{Hq}^{(1)})_{1,3} - 0.18 \, (c_{Hq}^{(1)})_{2,3} - 0.13 \, (c_{Hq}^{(3)})_{1,3} - 0.025 \, (c_{Hq}^{(3)})_{2,2} \\
& - [0.55 + 0.01i] \, (c_{Hq}^{(3)})_{2,3} + 0.02 \, (c_{Hq}^{(3)})_{3,3} - 0.39 \, (c_{Hud})_{3,3} - [0.58 + 0.01i] \, (c_{quqd}^{(1)})_{2,2,2,3} \\
& + 1.2 \, (c_{quqd}^{(1)})_{2,3,3,3} + [0.096 + 0.002i] \, (c_{quqd}^{(8)})_{2,2,2,3} - 0.20 (c_{quqd}^{(8)})_{2,3,3,3}. \tag{32}
\end{aligned}
$$

As mentioned above we can use the tree level dictionary to completely classify the models that induce a sizeable value for $C_{8gq}$ from running of tree level generated operators but we prefer to focus here on the direct one loop contribution to $C_{dG}$ taking full advantage of the part of the one loop dictionary computed in this work. We will therefore stick to models that do not give any tree level contribution to the SMEFT operators and consider the simpler case of

$$
\begin{aligned}
C_{8gd} &\approx [-6.9 + 2.9i] \times 10^3 \, (c_{dG})_{1,3} + [1.6 - 0.66i] \times 10^3 \, (c_{dG})_{2,3} - 65.5 \, (c_{dG})_{3,3} \\
&= [-27.6 + 11.6i] \times 10^3 \, (C_{dG})_{1,3} + [6.4 - 2.64i] \times 10^3 \, (C_{dG})_{2,3} - 262. \, (C_{dG})_{3,3}, \tag{33} \\
C_{8gs} &\approx [373 + 7i] \, (c_{dG})_{1,3} + [1600 + 30i] \, (c_{dG})_{2,3} - 65.5 \, (c_{dG})_{3,3} \\
&= [1492 + 28i] \, (C_{dG})_{1,3} + [6400 + 120i] \, (C_{dG})_{2,3} - 262. \, (C_{dG})_{3,3}, \tag{34}
\end{aligned}
$$

where in the second line of each equation we have gone back to dimensionful WC (arbitrary $\Lambda$) and all dimensionful quantities are measured in TeV. There is a continuum of solutions for these observables to match the benchmark values. An example of such solution is

$$(C_{dG})_{1,3} \approx -(1.1 + 0.3i) \times 10^{-5}, \quad (C_{dG})_{2,3} \approx -1 \times 10^{-5}, \quad (C_{dG})_{3,3} \approx 10^{-4}, \tag{35}$$

all in units of TeV$^{-2}$ and the complex value for the 1,3 entry is just to accommodate the assumption of real WC made in [26].

It is easy to generate these values in phenomenologically viable models. We have described in the previous section how to use SOLD to list all the models that generate the $C_{dG}$ WC in the Warsaw basis. From the (long) list we eliminate the cases in which at least one heavy field has

all its quantum numbers fixed as they correspond to linear couplings to the SM and therefore they have tree level contributions to other operators.

All the remaining models have three heavy fields. We choose one of the simplest ones that has contributions not suppressed by the down-type quark Yukawa couplings (for simplicity we use the conjugated field of the first fermion with respect to the one given by SOLD):

$$\Phi \sim (1,1)_{Y_\Phi}, \quad \Psi_1 \sim (3,2)_{\frac{1}{6}-Y_\Phi}, \quad \Psi_2 \sim (3,1)_{-\frac{1}{3}-Y_\Phi}, \tag{36}$$

where the hypercharge of the heavy scalar $Y_\Phi$ is arbitrary up to the limitation of no tree level contributions that restricts $Y_\Phi \neq 0, -1$. The complete expression is provided by the command

```
In[10]:= Match2Warsaw[alphaOdG[i,j],{Sa ->{1,1,Y1},Fa->{3,2,1/6-Y1},
         Fb->{3,1,-1/3-Y1}}]
```

For simplicity we reproduce it in the large scalar mass limit and neglecting terms suppressed by the down-type quark masses, that reads

$$
\begin{aligned}
(C_{dG})_{ij} = -\frac{g_3}{64\pi^2} & \frac{1}{M_\Phi^2(M_a^2 - M_b^2)}\Bigg[ 2M_a M_b \log\left(\frac{M_a^2}{M_b^2}\right)\lambda_{ab}^R \\
& + \left( 2M_b^2 \log\left(\frac{M_a^2}{M_b^2}\right) + (M_a^2 - M_b^2)\left(3 + 2\log\left(\frac{M_a^2}{M_\Phi^2}\right)\right)\right)\lambda_{ab}^L \Bigg](\lambda_{qa})_i (\lambda_{bd})_j + \mathcal{O}\left(\frac{M_{a,b}^2}{M_\Phi^4}\right),
\end{aligned}
\tag{37}
$$

where the masses and couplings are defined by the following Lagrangian

$$
\begin{aligned}
\mathcal{L} = & -M_\Phi^2 \Phi^\dagger \Phi - M_a \bar{\Psi}_a \Psi_a - M_b \bar{\Psi}_b \Psi_b \\
& - \bar{\Psi}_a [\lambda_{ab}^L P_L + \lambda_{ab}^R P_R]\Psi_b - (\lambda_{qa})_i \bar{q}_i P_R \Psi_a \Phi - (\lambda_{bd})_i \bar{\Psi}_b P_R d_i \Phi^\dagger + \dots,
\end{aligned}
\tag{38}
$$

with $P_{L,R} = (1 \mp \gamma^5)/2$ the chirality projectors.

Using the full expression given by SOLD we obtain that the following values of the parameters give the values for the WC in Eq.(35),

$$M_a = 1.5 \text{ TeV}, \quad M_b = 2.0 \text{ TeV}, \quad M_a = 4.0 \text{ TeV},$$
$$\lambda_{ab}^L = 0, \quad (\lambda_{qa})_i = \frac{(0.084 + 0.023i, 0.077, -0.776)}{\lambda_{ab}^R(\lambda_{bc})_3}. \tag{39}$$

We have then proceeded to perform the full one loop matching with matchmakereft via the function CompleteOneLoopMatching of this model. The result has been exported to WCxf format [34] and smelli [35–37] has been used to check the viability of the model. Indeed the following values of the remaining parameters

$$\lambda_{ab}^R = -0.7, \quad (\lambda_{bc})_3 = 0.9, \tag{40}$$

relax the corresponding tension for the considered operators without conflicting with other experimental observables (the global pull with respect to the SM is of 3.4 $\sigma$ when considering all other relevant observables encoded in smelli).

Note that, given the choice of quantum numbers, there is no linear coupling of the new fields to SM particles. This means that the lightest one, $\Psi_a$ in our case, is stable. We can make a choice of $Y_\Phi$ that ensures that $\Psi_a$ can still decay via higher dimensional operators, with a non-standard decay pattern, making it evade current experimental limits. See [38] for a detailed discussion.

# 6 Conclusions and Outlook

Effective field theories offer a new guiding principle in the quest of searching for new physics. The bottom-up approach provides an essentially model-independent parametrisation of experimental data and the top-down approach gives us the opportunity, via IR/UV dictionaries, to completely classify models of new physics with observable consequences. These dictionaries consist of a complete classification of new physics models that contribute to the effective Lagrangian at a certain order in the loop and operator dimension expansion, together with the explicit calculation of the WCs that each of these models generate. In this way, assuming that the dictionaries are computed up to the relevant order in the double (loop and operator dimension) perturbative expansion, we can use them to obtain in a systematic way the complete phenomenological implications of *any* observable model of new physics.

In this work we have reported the first step towards the calculation of the IR/UV dictionary for the SMEFT at mass dimension 6 and one loop order. In particular we have completely classified the most general renormalisable extension of the SM with new scalar or fermion fields that contribute to operators in the Warsaw basis containing at least one gauge field-strength tensor. These operators are the only ones in the Warsaw basis that cannot be generated at tree level in weakly coupled extensions of the SM and therefore the one loop contribution we have computed is the leading one for them. The complete list of models is of infinite length but it can be given in a closed form as a finite number of conditions on the quantum numbers of the new fields. Together with this classification we have also computed the WCs of the operators we are interested in for any of the allowed extensions. All these results have been encoded in the `Mathematica` package SOLD (SMEFT One Loop Dictionary), which relies heavily on `GroupMath` [23] for the group theoretic calculations. Since, for a particular SM extension, the user can be also interested in the WCs of the remaining operators in the Warsaw basis, SOLD has functions to call `matchmakereft` so that the complete tree level and one loop matching can be performed in a fully automated way for any model of interest (in practice for any model that is an extension of the SM with an arbitrary number of heavy scalar or fermion fields in any gauge representation). Incidentally, we have found that all the operators in our list can indeed be generated at the one loop order with the exception of the CP-violating operators with three field-strength tensors, which can only be generated, at least in the extensions considered, at the two loop order.

We have shown how to use the dictionary contained in SOLD which, despite the fact that is only a first step towards the complete dictionary, is already very useful for phenomenological studies. Nevertheless we plan to extend it in the near future to obtain the complete one loop dictionary at dimension 6 for the SMEFT. The next steps are the inclusion of heavy vectors and non-renormalisable interactions and the consideration of the remaining operators in the Warsaw basis, operators that can be potentially generated at tree level in weakly coupled extensions of the SM. All these results will be incorporated in SOLD. Future versions of `matchmakereft` will include a `Mathematica` interface to the `MatchingDB` format [39] that will allow for a flexible use of these highly non-trivial dictionaries.

## Acknowledgments

We thank J.C. Criado, R. Fonseca and J. Fuentes-Martín for useful discussions and especially M. Chala for useful discussions and comments on the manuscript. JS would like to thank the organisers of La Thuile 2023 for a nice atmosphere while this work was being finished and especially to U. Haisch, G. Isidori, M. Neubert and A.E. Thomsen for useful comments. We would like to thank Javier Olgoso for the design of the logo.

**Funding information**   This work has been partially supported by the Ministry of Science and Innovation and SRA (10.13039/501100011033) under grant PID2019-106087GB-C22, by the Junta de Andalucía grants FQM 101 and P18-FR-4314 (FEDER) and by the Deutsche Forschungsgemeinschaft (DFG, German Research Foundation) under grant 491245950 and under Germany's Excellence Strategy – EXC 2121 "Quantum Universe" – 390833306. PO is funded by an FPU grant from the Spanish government.

# A   Complete results for the $X^3$ class

The contributions for the $X^3$ class are simple enough that the complete classification and even the full result can be given in closed form. We can define the following operator:

$$\mathcal{O}_{3V} = \alpha_{3V} f^{ABC} V_\mu^{A\nu} V_\nu^{B\rho} V_\rho^{C\mu}, \tag{A.1}$$

for a general (non-abelian) gauge symmetry, with $f^{ABC}$ the structure constants of the group. This allows us to give the results for both $\alpha_{3W}$ and $\alpha_{3G}$ in the SMEFT. The only restriction on the heavy fields is that they are charged under the gauge symmetry, irrespectively on their hypercharge. The matching condition is the following [40]:

$$\alpha_{3V} = -\frac{1}{(4\pi)^2} \sum_R \frac{c_R\, g^3}{90 M_R^2} \mu(R), \qquad c_R = \begin{cases} 1, & \text{Dirac fermions,} \\ \frac{1}{2}, & \text{Majorana fermions,} \\ -\frac{1}{2}, & \text{complex scalars,} \\ -\frac{1}{4}, & \text{real scalars,} \end{cases} \tag{A.2}$$

with $\text{Tr}(T_R^A T_R^B) = \mu(R)\delta^{AB}$ where $R$ runs over all the heavy fields in the model, $T_R$ are the generators of the group in $R$'s representation, $g$ is the gauge group's coupling constant.

# B   Evanescent contribution to the dipole operators

We discuss in this section the one loop effects induced by the tree level generation of evanescent operators. This has been studied in detail in [22] with the result that, among the three classes of operators considered in this work, only the dipole operators receive a contribution from evanescent ones. The shifts in the dipole operators, as computed in [22], using the conventions in [1], read:

$$(\alpha_{eB})_{ij} \to (\alpha_{eB})_{ij} + \frac{g_1}{(4\pi)^2}\left[\frac{3}{8}(\gamma_{le})_{ijst}(Y_e)_{ts} - \frac{5}{8}(1-\xi_{rp})(Y_u)_{ts}^*(\gamma_{uelq}^c)_{sjit} \right.$$
$$\left. +\frac{5}{8}(1-\xi_{rp})(\gamma_{luqe})_{itsj}(Y_u)_{st}^* \right], \tag{B.1}$$

$$(\alpha_{eW})_{ij} \to (\alpha_{eW})_{ij} + \frac{g_2}{(4\pi)^2}\left[-\frac{1}{8}(Y_e)_{ts}(\gamma_{le})_{ijst} + \frac{3}{8}(1-\xi_{rp})(Y_u)_{ts}^*(\gamma_{uelq}^c)_{sjit} \right.$$
$$\left. -\frac{3}{8}(1-\xi_{rp})(Y_u)_{st}^*(\gamma_{luqe})_{itsj} \right], \tag{B.2}$$

$$(\alpha_{uB})_{ij} \to (\alpha_{uB})_{ij} - \frac{g_1}{(4\pi)^2}\frac{5}{8}(Y_u)_{ts}(\gamma_{qu})_{ijst}, \tag{B.3}$$

$$(\alpha_{uW})_{ij} \to (\alpha_{uW})_{ij} - \frac{g_2}{(4\pi)^2}\frac{3}{8}(Y_u)_{ts}(\gamma_{qu})_{ijst}, \tag{B.4}$$

$$(\alpha_{uG})_{ij} \to (\alpha_{uG})_{ij} - \frac{g_3}{(4\pi)^2}\frac{1}{4}(Y_u)_{ts}(\gamma_{qu}^{(8)})_{ijst}, \tag{B.5}$$

$$(\alpha_{dB})_{ij} \rightarrow (\alpha_{dB})_{ij} + \frac{g_1}{(4\pi)^2}\frac{1}{8}(Y_d)_{ts}(\gamma_{qd})_{ijst}\,, \tag{B.6}$$

$$(\alpha_{dW})_{ij} \rightarrow (\alpha_{dW})_{ij} - \frac{g_2}{(4\pi)^2}\frac{3}{8}(Y_d)_{ts}(\gamma_{qd})_{ijst}\,, \tag{B.7}$$

$$(\alpha_{dG})_{ij} \rightarrow (\alpha_{dG})_{ij} - \frac{g_3}{(4\pi)^2}\frac{1}{4}(Y_d)_{ts}(\gamma_{qd}^{(8)})_{ijst}\,. \tag{B.8}$$

In the equations above $Y_{u,d,e}$ stand for the up-type, down-type and charged electron Yukawa couplings, respectively, and $\xi_{rp}$ represents a reading point parameter and has xRP as output format in SOLD. More information on this parameter can be found in [22]. The remaining coefficients correspond to tree level contributions to evanescent structures. Using the notation in the tree level dictionary [4] they correspond to the following expressions

$$(\gamma_{le})_{ijkl} = \frac{(y_\varphi^e)_{ji}^*(y_\varphi^e)_{kl}}{M_\varphi^2}\,, \tag{B.9}$$

$$(\gamma_{uelq}^c)_{ijkl} = -\frac{(y_{\omega_1}^{eu})_{ji}(y_{\omega_1}^{ql})_{lk}^*}{M_{\omega_1}^2}\,, \tag{B.10}$$

$$(\gamma_{luqe})_{ijkl} = \frac{(y_{\Pi_7}^{lu})_{ij}(y_{\Pi_7}^{eq})_{lk}^*}{M_{\Pi_7}^2}\,, \tag{B.11}$$

$$(\gamma_{qu})_{ijkl} = \frac{(y_\varphi^u)_{ij}(y_\varphi^u)_{lk}^*}{M_\varphi^2}\,, \tag{B.12}$$

$$(\gamma_{qd})_{ijkl} = \frac{(y_\varphi^d)_{ji}^*(y_\varphi^d)_{kl}}{M_\varphi^2}\,, \tag{B.13}$$

$$(\gamma_{qu}^{(8)})_{ijkl} = \frac{(y_\Phi^u)_{ij}(y_\Phi^u)_{lk}^*}{M_\Phi^2}\,, \tag{B.14}$$

$$(\gamma_{qd}^{(8)})_{ijkl} = \frac{(y_\Phi^d)_{ji}^*(y_\Phi^d)_{kl}}{M_\Phi^2}\,. \tag{B.15}$$



## C  Redundant and evanescent operators

In this appendix we list the redundant (Table 2) and evanescent (Table 3) operators that are relevant for the calculation of the physical operators considered in this work.

Table 2:  One loop generated redundant operators. Operators in gray do not contribute to one loop generated operators in the Warsaw basis. Shaded operators are generated at two or higher order loops in SM extensions with fermions and scalars.

| $\psi^2 D^3$ | | $\psi^2 XD$ | | | |
|---|---|---|---|---|---|
| $\mathcal{R}_{qD}$ | $\frac{i}{2}\overline{q}\{D_\mu D^\mu,\slashed{D}\}q$ | $\mathcal{R}_{Gq}$ | $(\overline{q}T^A\gamma^\mu q)D^\nu G^A_{\mu\nu}$ | $\mathcal{R}_{Bd}$ | $(\overline{d}\gamma^\mu d)\partial^\nu B_{\mu\nu}$ |
| $\mathcal{R}_{uD}$ | $\frac{i}{2}\overline{u}\{D_\mu D^\mu,\slashed{D}\}u$ | $\mathcal{R}'_{Gq}$ | $\frac{1}{2}(\overline{q}T^A\gamma^\mu i\overleftrightarrow{D}^\nu q)G^A_{\mu\nu}$ | $\mathcal{R}'_{Bd}$ | $\frac{1}{2}(\overline{d}\gamma^\mu i\overleftrightarrow{D}^\nu d)B_{\mu\nu}$ |
| $\mathcal{R}_{dD}$ | $\frac{i}{2}\overline{d}\{D_\mu D^\mu,\slashed{D}\}d$ | $\mathcal{R}'_{\widetilde{G}q}$ | $\frac{1}{2}(\overline{q}T^A\gamma^\mu i\overleftrightarrow{D}^\nu q)\widetilde{G}^A_{\mu\nu}$ | $\mathcal{R}'_{\widetilde{B}d}$ | $\frac{1}{2}(\overline{d}\gamma^\mu i\overleftrightarrow{D}^\nu d)\widetilde{B}_{\mu\nu}$ |
| $\mathcal{R}_{\ell D}$ | $\frac{i}{2}\overline{\ell}\{D_\mu D^\mu,\slashed{D}\}\ell$ | $\mathcal{R}_{Wq}$ | $(\overline{q}\sigma^I\gamma^\mu q)D^\nu W^I_{\mu\nu}$ | $\mathcal{R}_{W\ell}$ | $(\overline{\ell}\sigma^I\gamma^\mu\ell)D^\nu W^I_{\mu\nu}$ |
| $\mathcal{R}_{eD}$ | $\frac{i}{2}\overline{e}\{D_\mu D^\mu,\slashed{D}\}e$ | $\mathcal{R}'_{Wq}$ | $\frac{1}{2}(\overline{q}\sigma^I\gamma^\mu i\overleftrightarrow{D}^\nu q)W^I_{\mu\nu}$ | $\mathcal{R}'_{W\ell}$ | $\frac{1}{2}(\overline{\ell}\sigma^I\gamma^\mu i\overleftrightarrow{D}^\nu\ell)W^I_{\mu\nu}$ |
| $\psi^2 HD^2$ + **h.c.** | | $\mathcal{R}'_{\widetilde{W}q}$ | $\frac{1}{2}(\overline{q}\sigma^I\gamma^\mu i\overleftrightarrow{D}^\nu q)\widetilde{W}^I_{\mu\nu}$ | $\mathcal{R}'_{\widetilde{W}\ell}$ | $\frac{1}{2}(\overline{\ell}\sigma^I\gamma^\mu i\overleftrightarrow{D}^\nu\ell)\widetilde{W}^I_{\mu\nu}$ |
| $\mathcal{R}_{uHD1}$ | $(\overline{q}u)D_\mu D^\mu\widetilde{H}$ | $\mathcal{R}_{Bq}$ | $(\overline{q}\gamma^\mu q)\partial^\nu B_{\mu\nu}$ | $\mathcal{R}_{B\ell}$ | $(\overline{\ell}\gamma^\mu\ell)\partial^\nu B_{\mu\nu}$ |
| $\mathcal{R}_{uHD2}$ | $(\overline{q}\,i\sigma_{\mu\nu}D^\mu u)D^\nu\widetilde{H}$ | $\mathcal{R}'_{Bq}$ | $\frac{1}{2}(\overline{q}\gamma^\mu i\overleftrightarrow{D}^\nu q)B_{\mu\nu}$ | $\mathcal{R}'_{B\ell}$ | $\frac{1}{2}(\overline{\ell}\gamma^\mu i\overleftrightarrow{D}^\nu\ell)B_{\mu\nu}$ |
| $\mathcal{R}_{uHD3}$ | $(\overline{q}D_\mu D^\mu u)\widetilde{H}$ | $\mathcal{R}'_{\widetilde{B}q}$ | $\frac{1}{2}(\overline{q}\gamma^\mu i\overleftrightarrow{D}^\nu q)\widetilde{B}_{\mu\nu}$ | $\mathcal{R}'_{\widetilde{B}\ell}$ | $\frac{1}{2}(\overline{\ell}\gamma^\mu i\overleftrightarrow{D}^\nu\ell)\widetilde{B}_{\mu\nu}$ |
| $\mathcal{R}_{uHD4}$ | $(\overline{q}D_\mu u)D^\mu\widetilde{H}$ | $\mathcal{R}_{Gu}$ | $(\overline{u}T^A\gamma^\mu u)D^\nu G^A_{\mu\nu}$ | $\mathcal{R}_{Be}$ | $(\overline{e}\gamma^\mu e)\partial^\nu B_{\mu\nu}$ |
| $\mathcal{R}_{dHD1}$ | $(\overline{q}d)D_\mu D^\mu H$ | $\mathcal{R}'_{Gu}$ | $\frac{1}{2}(\overline{u}T^A\gamma^\mu i\overleftrightarrow{D}^\nu u)G^A_{\mu\nu}$ | $\mathcal{R}'_{Be}$ | $\frac{1}{2}(\overline{e}\gamma^\mu i\overleftrightarrow{D}^\nu e)B_{\mu\nu}$ |
| $\mathcal{R}_{dHD2}$ | $(\overline{q}\,i\sigma_{\mu\nu}D^\mu d)D^\nu H$ | $\mathcal{R}'_{\widetilde{G}u}$ | $\frac{1}{2}(\overline{u}T^A\gamma^\mu i\overleftrightarrow{D}^\nu u)\widetilde{G}^A_{\mu\nu}$ | $\mathcal{R}'_{\widetilde{B}e}$ | $\frac{1}{2}(\overline{e}\gamma^\mu i\overleftrightarrow{D}^\nu e)\widetilde{B}_{\mu\nu}$ |
| $\mathcal{R}_{dHD3}$ | $(\overline{q}D_\mu D^\mu d)H$ | $\mathcal{R}_{Bu}$ | $(\overline{u}\gamma^\mu u)\partial^\nu B_{\mu\nu}$ | | |
| $\mathcal{R}_{dHD4}$ | $(\overline{q}D_\mu d)D^\mu H$ | $\mathcal{R}'_{Bu}$ | $\frac{1}{2}(\overline{u}\gamma^\mu i\overleftrightarrow{D}^\nu u)B_{\mu\nu}$ | | |
| $\mathcal{R}_{eHD1}$ | $(\overline{\ell}e)D_\mu D^\mu H$ | $\mathcal{R}'_{\widetilde{B}u}$ | $\frac{1}{2}(\overline{u}\gamma^\mu i\overleftrightarrow{D}^\nu u)\widetilde{B}_{\mu\nu}$ | | |
| $\mathcal{R}_{eHD2}$ | $(\overline{\ell}\,i\sigma_{\mu\nu}D^\mu e)D^\nu H$ | $\mathcal{R}_{Gd}$ | $(\overline{d}T^A\gamma^\mu d)D^\nu G^A_{\mu\nu}$ | | |
| $\mathcal{R}_{eHD3}$ | $(\overline{\ell}D_\mu D^\mu e)H$ | $\mathcal{R}'_{Gd}$ | $\frac{1}{2}(\overline{d}T^A\gamma^\mu i\overleftrightarrow{D}^\nu d)G^A_{\mu\nu}$ | | |
| $\mathcal{R}_{eHD4}$ | $(\overline{\ell}D_\mu e)D^\mu H$ | $\mathcal{R}'_{\widetilde{G}d}$ | $\frac{1}{2}(\overline{d}T^A\gamma^\mu i\overleftrightarrow{D}^\nu d)\widetilde{G}^A_{\mu\nu}$ | | |

Table 3: One loop generated evanescent operators. Operators in gray do not contribute to one loop generated operators in the Warsaw basis. Shaded operators are generated at two or higher order loops in SM extensions with fermions and scalars.

| $\Psi^2XH$ + h.c. | | $\Psi^2XD$ | | | |
|---|---|---|---|---|---|
| $\mathcal{E}_{uG}$ | $\bar{q}T^A\sigma^{\mu\nu}u\widetilde{H}\widetilde{G}^A_{\mu\nu}$ | $\mathcal{E}_{Gq}$ | $\bar{q}T^A(\sigma^{\mu\nu}\gamma^\rho + \gamma^\rho\sigma^{\mu\nu})qD_\rho\widetilde{G}^A_{\mu\nu}$ | $\mathcal{E}_{Gd}$ | $\bar{d}T^A(\sigma^{\mu\nu}\gamma^\rho + \gamma^\rho\sigma^{\mu\nu})dD_\rho\widetilde{G}^A_{\mu\nu}$ |
| $\mathcal{E}_{uW}$ | $\bar{q}\sigma^I\sigma^{\mu\nu}u\widetilde{H}\widetilde{W}^I_{\mu\nu}$ | $\mathcal{E}'_{Gq}$ | $i\bar{q}(T^A\sigma^{\mu\nu}\slashed{D} - \overleftarrow{\slashed{D}}\sigma^{\mu\nu}T^A)qG^A_{\mu\nu}$ | $\mathcal{E}'_{Gd}$ | $i\bar{d}(T^A\sigma^{\mu\nu}\slashed{D} - \overleftarrow{\slashed{D}}\sigma^{\mu\nu}T^A)dG^A_{\mu\nu}$ |
| $\mathcal{E}_{uB}$ | $\bar{q}\sigma^{\mu\nu}u\widetilde{H}\widetilde{B}_{\mu\nu}$ | $\mathcal{E}'_{\widetilde{G}q}$ | $i\bar{q}(T^A\sigma^{\mu\nu}\slashed{D} - \overleftarrow{\slashed{D}}\sigma^{\mu\nu}T^A)q\widetilde{G}^A_{\mu\nu}$ | $\mathcal{E}'_{\widetilde{G}d}$ | $i\bar{d}(T^A\sigma^{\mu\nu}\slashed{D} - \overleftarrow{\slashed{D}}\sigma^{\mu\nu}T^A)d\widetilde{G}^A_{\mu\nu}$ |
| $\mathcal{E}_{dG}$ | $\bar{q}T^A\sigma^{\mu\nu}dH\widetilde{G}^A_{\mu\nu}$ | $\mathcal{E}_{Wq}$ | $\bar{q}\sigma^I(\sigma^{\mu\nu}\gamma^\rho + \gamma^\rho\sigma^{\mu\nu})qD_\rho\widetilde{W}^I_{\mu\nu}$ | $\mathcal{E}_{Bd}$ | $\bar{d}(\sigma^{\mu\nu}\gamma^\rho + \gamma^\rho\sigma^{\mu\nu})d\partial_\rho\widetilde{B}_{\mu\nu}$ |
| $\mathcal{E}_{dW}$ | $\bar{q}\sigma^I\sigma^{\mu\nu}dH\widetilde{W}^I_{\mu\nu}$ | $\mathcal{E}'_{Wq}$ | $i\bar{q}(\sigma^I\sigma^{\mu\nu}\slashed{D} - \overleftarrow{\slashed{D}}\sigma^{\mu\nu}\sigma^I)qW^I_{\mu\nu}$ | $\mathcal{E}'_{Bd}$ | $i\bar{d}(\sigma^{\mu\nu}\slashed{D} - \overleftarrow{\slashed{D}}\sigma^{\mu\nu})dB^A_{\mu\nu}$ |
| $\mathcal{E}_{dB}$ | $\bar{q}\sigma^{\mu\nu}dH\widetilde{B}_{\mu\nu}$ | $\mathcal{E}'_{\widetilde{W}q}$ | $i\bar{q}(\sigma^I\sigma^{\mu\nu}\slashed{D} - \overleftarrow{\slashed{D}}\sigma^{\mu\nu}\sigma^I)q\widetilde{W}^I_{\mu\nu}$ | $\mathcal{E}'_{\widetilde{B}d}$ | $i\bar{d}(\sigma^{\mu\nu}\slashed{D} - \overleftarrow{\slashed{D}}\sigma^{\mu\nu})d\widetilde{B}_{\mu\nu}$ |
| $\mathcal{E}_{eW}$ | $\bar{\ell}\sigma^I\sigma^{\mu\nu}eH\widetilde{W}^I_{\mu\nu}$ | $\mathcal{E}_{Bq}$ | $\bar{q}(\sigma^{\mu\nu}\gamma^\rho + \gamma^\rho\sigma^{\mu\nu})q\partial_\rho\widetilde{B}_{\mu\nu}$ | $\mathcal{E}_{W\ell}$ | $\bar{\ell}\sigma^I(\sigma^{\mu\nu}\gamma^\rho + \gamma^\rho\sigma^{\mu\nu})\ell D_\rho\widetilde{W}^I_{\mu\nu}$ |
| $\mathcal{E}_{eB}$ | $\bar{\ell}\sigma^{\mu\nu}eH\widetilde{B}_{\mu\nu}$ | $\mathcal{E}'_{Bq}$ | $i\bar{q}(\sigma^{\mu\nu}\slashed{D} - \overleftarrow{\slashed{D}}\sigma^{\mu\nu})qB_{\mu\nu}$ | $\mathcal{E}'_{W\ell}$ | $i\bar{\ell}(\sigma^I\sigma^{\mu\nu}\slashed{D} - \overleftarrow{\slashed{D}}\sigma^{\mu\nu}\sigma^I)\ell W^I_{\mu\nu}$ |
| $\psi^2HD^2$ + h.c. | | $\mathcal{E}'_{\widetilde{B}q}$ | $i\bar{q}(\sigma^{\mu\nu}\slashed{D} - \overleftarrow{\slashed{D}}\sigma^{\mu\nu})q\widetilde{B}_{\mu\nu}$ | $\mathcal{E}'_{\widetilde{W}\ell}$ | $i\bar{\ell}(\sigma^I\sigma^{\mu\nu}\slashed{D} - \overleftarrow{\slashed{D}}\sigma^{\mu\nu}\sigma^I)\ell\widetilde{W}^I_{\mu\nu}$ |
| $\mathcal{E}_{uH}$ | $\bar{q}\sigma^{\mu\nu}D^\rho u D^\sigma\widetilde{H}\epsilon_{\mu\nu\rho\sigma}$ | $\mathcal{E}_{Gu}$ | $\bar{u}T^A(\sigma^{\mu\nu}\gamma^\rho + \gamma^\rho\sigma^{\mu\nu})uD_\rho\widetilde{G}^A_{\mu\nu}$ | $\mathcal{E}_{B\ell}$ | $\bar{\ell}(\sigma^{\mu\nu}\gamma^\rho + \gamma^\rho\sigma^{\mu\nu})\ell\partial_\rho\widetilde{B}_{\mu\nu}$ |
| $\mathcal{E}_{dH}$ | $\bar{q}\sigma^{\mu\nu}D^\rho d D^\sigma H\epsilon_{\mu\nu\rho\sigma}$ | $\mathcal{E}'_{Gu}$ | $i\bar{u}(T^A\sigma^{\mu\nu}\slashed{D} - \overleftarrow{\slashed{D}}\sigma^{\mu\nu}T^A)uG^A_{\mu\nu}$ | $\mathcal{E}'_{B\ell}$ | $i\bar{\ell}(\sigma^{\mu\nu}\slashed{D} - \overleftarrow{\slashed{D}}\sigma^{\mu\nu})\ell B_{\mu\nu}$ |
| $\mathcal{E}_{eH}$ | $\bar{\ell}\sigma^{\mu\nu}D^\rho e D^\sigma H\epsilon_{\mu\nu\rho\sigma}$ | $\mathcal{E}'_{\widetilde{G}u}$ | $i\bar{u}(T^A\sigma^{\mu\nu}\slashed{D} - \overleftarrow{\slashed{D}}\sigma^{\mu\nu}T^A)u\widetilde{G}^A_{\mu\nu}$ | $\mathcal{E}'_{\widetilde{B}\ell}$ | $i\bar{\ell}(\sigma^{\mu\nu}\slashed{D} - \overleftarrow{\slashed{D}}\sigma^{\mu\nu})\ell\widetilde{B}_{\mu\nu}$ |
| | | $\mathcal{E}_{Bu}$ | $\bar{u}(\sigma^{\mu\nu}\gamma^\rho + \gamma^\rho\sigma^{\mu\nu})u\partial_\rho\widetilde{B}_{\mu\nu}$ | $\mathcal{E}_{Be}$ | $\bar{e}(\sigma^{\mu\nu}\gamma^\rho + \gamma^\rho\sigma^{\mu\nu})e\partial_\rho\widetilde{B}_{\mu\nu}$ |
| | | $\mathcal{E}'_{Bu}$ | $i\bar{u}(\sigma^{\mu\nu}\slashed{D} - \overleftarrow{\slashed{D}}\sigma^{\mu\nu})uB_{\mu\nu}$ | $\mathcal{E}'_{Be}$ | $i\bar{e}(\sigma^{\mu\nu}\slashed{D} - \overleftarrow{\slashed{D}}\sigma^{\mu\nu})eB_{\mu\nu}$ |
| | | $\mathcal{E}'_{\widetilde{B}u}$ | $i\bar{u}(\sigma^{\mu\nu}\slashed{D} - \overleftarrow{\slashed{D}}\sigma^{\mu\nu})u\widetilde{B}_{\mu\nu}$ | $\mathcal{E}'_{\widetilde{B}e}$ | $i\bar{e}(\sigma^{\mu\nu}\slashed{D} - \overleftarrow{\slashed{D}}\sigma^{\mu\nu})e\widetilde{B}_{\mu\nu}$ |

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
