# Peer review of "Towards the one loop IR/UV dictionary in the SMEFT: one loop generated operators from new scalars and fermions"

_SciPost Physics, doi:SciPost Phys. 15, 143 (2023)_

## Round 1 · Referee Report · Anonymous (Referee 1) · 2023-6-6

Report

Dear Editor,

In this manuscript, the author presents a Mathematica package, called \texttt{SOLD}, that offers a complete set of IR/UV dictionaries for operators in the Standard Model Effective Field Theories (SMEFT) that are only generated at the one-loop level. The study is motivated by the possible loop-level sensitivity of forthcoming experiments for certain UV models and our ultimate goal to gain a deeper understanding of the UV origin of the Standard Model.

The authors begin by outlining a general method for classifying the UV origin at the one-loop order. They achieve this by identifying and analyzing all the vertices present in the one-light-particle-irreducible amplitudes responsible for generating the target operator. To put this method into practice, the author constructs a highly generic UV model that encompasses all possible renormalizable interactions for fermions and bosons. They then utilize their previously developed automated matching tools to determine which interactions correspond to specific operators. Afterward, the authors employ the package GroupMath to derive constraints on the gauge representations of new fields within the UV model. By doing so, they are able to condense an infinite number of concrete UV models into a finite set of conditions based on the quantum numbers of the new fields, which is an elegant result. Subsequently, the authors provide a detailed explanation of how to use each function in the \texttt{SOLD} package, offering concrete examples and instructions for obtaining results at each step. They also demonstrate how to extract further matching results and the associated Lagrangians of the relevant models. Finally, the authors include a pedagogical example for the practical application of their findings in a real phenomenological study.

Overall, this manuscript presents a valuable contribution to the phenomenological study in high energy physics. The \texttt{SOLD} package, along with the provided practical examples, equips researchers with powerful tools for exploring the UV origin of operators in the SMEFT at one-loop order. The clarity of the explanations and the inclusion of concrete instructions make this manuscript an accessible resource for both novice and expert in the field.

However, before publication, I wish authors could help to clarify the following questions:

  1. It seems that at the dimension six level, none of the operator containing field strength can be generated at the tree level (assuming the renormalizable UV interactions). Does this feature extend to higher dimensions?

  2. In the first paragraph in section 3.2, the authors say that an infinite number of representations can satisfy the restrictions derived from certain topologies. Can some high dimensional representation hurt the asymptotic free of QCD?

  3. I want to confirm my understanding. In the second paragraph in section 3.2, start from "Note however" to "between different diagrams". Does that mean for a concrete model that satisfies certain restrictions in the output of the function "ListModelsWarsaw", its contribution to the Wilson coefficient as the input argument of the "ListModelsWarsaw" function can still be zero?

  4. If the above answer is true, is that true that the output of "ListModelsWarsaw" gives an overcomplete list of the possible models and the only way to verify their contributions to certain operators is to perform the matching with the function "Match2Waswar"?

  5. In eq.3.2 the definition of the covariant derivative seems to miss the U(1) hypercharge for the fields in the second term.

  • validity: top
  • significance: high
  • originality: high
  • clarity: high
  • formatting: excellent
  • grammar: excellent

Author:  Pablo Olgoso  on 2023-06-23  [id 3755]

(in reply to Report 1 on 2023-06-06)

Dear Referee, we would like to thank you for your thorough review and your positive feedback. We hope our answers below clarify your questions.

  1. This feature does not, indeed, extend to higher dimensions. Operators, for instance, of the class $\psi^4 X_{\mu\nu}$ can be generated at tree level, as it was shown in Ref. [11] from our manuscript.
  2. Yes, high color representations can affect and crucially modify the strong coupling beta function at high energies.
  3. Yes, there can be cancellations that depend on the details of the specific UV model.
  4. This is indeed correct, but the number of these cases in practice is very limited.
  5. We will correct this typo in the updated version of the manuscript.

Best regards, the authors

---

## Round 1 · Referee Report · Joydeep Chakrabortty (Referee 2) · 2023-6-16

Report

In the manuscript titled "Towards the one loop IR/UV dictionary in the SMEFT: one loop generated operators from new scalars and fermions" authors have developed a Mathematica based package "SOLD" that enables to compute complete tree and 1-loop integrating out with the help of another previously developed program "matchmakereft". As recently EFT has drawn a lot of attention this kind of programs will certainly be useful.

I have a few comments on this manuscript:

  1. Is the out of effective operators restrcited to the WARSAW basis only? Or is it possible to get the operators in other basis, specifically Green's set as well?

  2. Can this program work for any arbitrary representation under Standard Model gauge group? Is it possible to include the information of discrete symmetry in an UV theory?

  3. How one can possibly use this program if the internal symmetry group (gauge) is extended? Or it works for SM gauge group only?

  4. It will be helpful for the users if the contributions from only heavy propagators, heavy-light mixed propagators can be extracted out. How the mixed statistics diagrams, i.e., in the loop the particles have different spins are evaluated in this program?

  5. Does this program work for majorana fermions or only vector like fermions? How the chirality operators are dealt with in the process of computing the loops? What is the regularization prescription used here?

I would like to know these facts before I recommend this paper for publication.

  • validity: -
  • significance: -
  • originality: -
  • clarity: -
  • formatting: -
  • grammar: -

Author:  Pablo Olgoso  on 2023-06-23  [id 3754]

(in reply to Report 2 by Joydeep Chakrabortty on 2023-06-16)

Dear Referee, we would like to first thank you for your positive feedback. We list below our clarifications to your comments:

  1. There are functions to perform all the relevant tasks both in the Green’s and physical basis. For instance, you can obtain the matching conditions using Match2Green or Match2Warsaw.
  2. Yes, arbitrary representations of the SM gauge group can be used in SOLD. Discrete symmetries are however not supported, but they can be implemented ‘’manually’’ by turning off the relevant couplings in the result.
  3. We present in our article the calculation of the dictionary for the SMEFT. Thus, the field content and symmetries are fixed to be the SM ones in the EFT. Only the SM gauge group is supported, so any additional symmetry must be implemented in its broken phase within SOLD. Of course, matchmakereft is more general than that bu this generality goes beyond the scope of the present calculation.
  4. Both heavy and heavy/light topologies contribute to the final answer and we do not distinguish them within SOLD. All loop contributions are computed using the standard rules for the calculation of Feynman diagrams.
  5. Yes, both Majorana and Dirac fermions are supported. However, SOLD automatically considers Majorana fermions for real representations of the gauge group and Dirac for complex ones. If the user wants to have a Dirac field in a real representation, they should use two degenerate Majorana fields to reproduce this case. We would like to thank the referee for pointing this out and we suggest to add a sentence to our manuscript specifying this case explicitly. Regarding $\gamma_5$, as explained in Section 3.1, we use a Naive Dimensional Regularization prescription for $\gamma_5$. The only possible ambiguities related to this treatment appear in boxes of the type $H^\dagger H X_{\mu\nu} \tilde{X}^{\mu\nu}$ and are fixed by imposing the reality of the Wilson Coefficient.

Best regards, the authors

---

## Round 2 · Referee Report · Anonymous (Referee 3) · 2023-7-18

Report

The authors have implemented all the suggestions. I am happy with the new version and accept it for the publication.

---

## Round 2 · List of Changes

-Fixed a typo in Eq. (3.2).
-Added a comment about the implementation of real Dirac fermions.

---

## Editorial Decision

published